# Tracking of Ubiquitin Signaling through 3.5 Billion Years of Combinatorial Conjugation

**DOI:** 10.3390/ijms25168671

**Published:** 2024-08-08

**Authors:** Alena N. Kaminskaya, Alena S. Evpak, Alexey A. Belogurov, Anna A. Kudriaeva

**Affiliations:** 1Shemyakin and Ovchinnikov Institute of Bioorganic Chemistry, Russian Academy of Sciences, 117997 Moscow, Russia; kaminskayaan@mail.ru (A.N.K.); alena.st97@gmail.com (A.S.E.); 2Department of Biological Chemistry, Russian University of Medicine, Ministry of Health of Russian Federation, 127473 Moscow, Russia

**Keywords:** ubiquitin, ubiquitin ligase, RING, HECT, evolution, domain structure, phylum, eukaryote, prokaryote, Archaea

## Abstract

Ubiquitination is an evolutionary, ancient system of post-translational modification of proteins that occurs through a cascade involving ubiquitin activation, transfer, and conjugation. The maturation of this system has followed two main pathways. The first is the conservation of a universal structural fold of ubiquitin and ubiquitin-like proteins, which are present in both Archaea and Bacteria, as well as in multicellular Eukaryotes. The second is the rise of the complexity of the superfamily of ligases, which conjugate ubiquitin-like proteins to substrates, in terms of an increase in the number of enzyme variants, greater variation in structural organization, and the diversification of their catalytic domains. Here, we examine the diversity of the ubiquitination system among different organisms, assessing the variety and conservation of the key domains of the ubiquitination enzymes and ubiquitin itself. Our data show that E2 ubiquitin-conjugating enzymes of metazoan phyla are highly conservative, whereas the homology of E3 ubiquitin ligases with human orthologues gradually decreases depending on “molecular clock” timing and evolutionary distance. Surprisingly, Chordata and Echinodermata, which diverged over 0.5 billion years ago during the Cambrian explosion, share almost the same homology with humans in the amino acid sequences of E3 ligases but not in their adaptor proteins. These observations may suggest that, firstly, the E2 superfamily already existed in its current form in the last common metazoan ancestor and was generally not affected by purifying selection in metazoans. Secondly, it may indicate convergent evolution of the ubiquitination system and highlight E3 adaptor proteins as the “upper deck” of the ubiquitination system, which plays a crucial role in chordate evolution.

## 1. Introduction

Post-translational modifications of proteins represent the most significant regulators of both the physical and physiological properties of protein molecules. They play a pivotal role in the functioning of all living organisms, regulating numerous biochemical processes, including enzymatic activity, interaction with other proteins, and the lifespan of the protein within the cell. Proteins can be modified by small molecules, such as phosphate, methyl, or acetyl groups, or by other protein molecules, such as the ubiquitin (Ub) protein [1,2,3,4].

Ubiquitination is a highly versatile post-translational modification of proteins that is involved in multiple biological processes, including protein degradation, proliferation, differentiation, endocytosis, regulation of transcription and translation, DNA repair, transport, and response to stress and pathogens [5,6]. The conjugation of ubiquitin to a substrate is accomplished by a complex system of enzymes, E1 (activating enzyme), E2 (conjugating enzyme), and E3 ubiquitin ligase [7]. There are numerous ways in which proteins can be modified by ubiquitin, including monoubiquitination (at one or more sites) and polyubiquitination, with different binding parameters and lengths of ubiquitin chains. The ε-amino groups of seven lysine residues (K6, K11, K27, K29, K33, K48, K63) and the N-terminal methionine in ubiquitin enable the formation of isopeptide bonds, resulting in the generation of distinct types of ubiquitin branching that determine the fate of substrates [8]. The parameters of the chains are extremely diverse, with the potential for either homogeneous (i.e., forming bonds via lysine at strictly defined positions) or heterogeneous (combining different types of bonds) configurations. The latter can branch through ubiquitination at several sites simultaneously. The physiological significance of the ubiquitination signal is thought to depend on the type of bond and the length of the ubiquitin chain. A distinct class of proteins contains a ubiquitin-like domain that is not cleaved. In addition to the formation of bonds between ubiquitin monomers, conjugation of ubiquitin with ubiquitin-like proteins is also possible, thereby further expanding the functionality of this type of post-translational modification.

In Eukaryotes, in addition to ubiquitin, approximately 14 families of ubiquitin-like proteins (Ubls) have been identified. These exhibit a relatively distant similarity to ubiquitin in the amino acid sequence yet are largely similar in tertiary structure and mechanisms of attachment to their protein substrates. Further levels of complexity are introduced by the formation of heterotypic chains with mixed linkage types of branched chains and the formation of hybrid Ub chains with different Ubls [9]. Additionally, post-translational modifications of ubiquitin itself occur through phosphorylation, acetylation, SUMOylation, and NEDDylation. Each type of linkage has its own three-dimensional topology, which provides interaction with linkage-type-specific effector proteins to fulfill their unique biological functions. The diversity of functions of the ubiquitination system has expanded, including an increasing range of substrates undergoing this post-translational modification. It has been demonstrated that Ub can be conjugated to bacterial lipopolysaccharides and eukaryotic and viral phospholipids [10].

The aim of this review is to examine the functional features, global domain organization, and possible evolutionary pathways of ubiquitin itself and the components of the ubiquitination system, from bacteria to multicellular eukaryotic organisms, over billions of years of evolutionary refinement.

## 2. Fifty Shades of Ubiquitin

Ubiquitin is a ubiquitous protein molecule present in all eukaryotic organisms that facilitates the labeling of substrates. It is a highly conserved protein comprising 76 amino acids. Structurally, ubiquitin adopts a compact globular β-sheet conformation known as the “ubiquitin fold”, characterized by a five-strand β-sheet (five antiparallel β-strands) and a single helix at the apex. It also has an open C-terminal tail that extends to participate in covalent binding to targets.

Ubiquitin-like proteins can be classified into two categories (Figure 1A): (i) ubiquitin-like modifiers (ULMs), which covalently attach to target molecules via their C-termini [7,11], and (ii) proteins containing a ubiquitin-like domain (UDPs), which are similar to ubiquitin in terms of sequence and structure [11], but do not form conjugates with other molecules. Instead, they function by forming protein complexes or binding to their adaptor proteins [12]. The question of which is the primary entity in terms of ULM evolution—UDP or ubiquitin—remains open. Reconstruction of the evolutionary history of ubiquitin-like proteins based on bacterial and archaeal genomes indicates that the Last Universal Common Ancestor (LUCA) possessed genes encoding Ubl-binding proteins that recognize RNA [13,14]. Furthermore, it is hypothesized that the ubiquitin-like β-capture fold originated in the LUCA as an RNA-binding domain [13,14]. Despite the universality of the ubiquitination system, the representation of ubiquitin and Ubls differs between the genomes of different groups of organisms.

Ubiquitin-like proteins also exhibit a conserved globular β-capture conformation analogous to that of Ub (Figure 1A). The ubiquitin-like protein family includes SUMO (small ubiquitin-like modifier), NEDD8 (neural precursor cell expressed, developmentally down-regulated 8), ATG8 (autophagy-related protein 8), ATG12 (autophagy-related protein 12), URM1 (ubiquitin-related modifier 1), UFM1 (ubiquitin-fold modifier 1), Ubl5/Hub1 (ubiquitin-like protein 5), FUB1 (few ubiquitin-like protein 1), FAT10 (leukocyte antigen F-associated 10), and ISG15 (interferon-stimulated gene 15) proteins. In addition, individual Ubls, including SUMO, ISG15, and NEDD8, can participate in modifying lysine residues of ubiquitin. This results in the formation of a novel type of ubiquitin branching, the mixed type, as previously demonstrated by David A. Pérez Berrocal and his colleagues [9]. Ubls have a comparable, though often more specialized, function in cellular regulation to that of Ub. Mono- or poly-SUMOylation is involved in the regulation of several nuclear processes, including gene expression, DNA damage response, RNA processing, cell cycle progression, and proteostasis. Additionally, SUMOylation is associated with the regulation of immunity, pluripotency, and stress responses [15]. Furthermore, poly-SUMO chains can act as a signal for Ub-dependent proteasomal degradation upon recognition by specialized E3 ligases that possess multiple SUMO-interacting motifs [16]. NEDD8 is most notable for its involvement in the ubiquitination of a multitude of substrates involved in various aspects of the cell cycle [17]. The ATG8 and ATG12 families of ubiquitin-like modifiers are known to be involved in autophagy [18]. It has been demonstrated that Hub1 binds to several spliceosomal proteins, including the HIND-containing splicing factors Snu66/SART-1 and PRP-38 [19]. The primary target of UFM1 is the ribosomal protein RPL26/uL24, a member of the 60S ribosomal subunit [20]. UFMylation results in a relaxation of tight closure at the ribosome–translocon junction, thus enabling cytosolic proteasomes and the ribosome-associated protein synthesis quality control apparatus to gain access to peptides synthesized on the endoplasmic reticulum with the aim of performing quality control of their synthesis [21].

Ubiquitin and ubiquitin-like proteins are present both in Eukaryotes and Prokaryotes. The Ubls of Eukaryotes, Bacteria and Archaea exhibit a high degree of similarity in terms of 3D structures (Figure 1A) and are significantly more variable in amino acid sequences (Figure 1B).

### 2.1. Ubiquitin and Ubls of Multicellular Eukaryotes

In the human genome, the ubiquitin gene is represented by four different genes, two of which, UBB (ubiquitin B) and UBC (ubiquitin C), encode polyubiquitin. The other two genes, UBA52 (ubiquitin A-52 residue ribosomal fusion protein) and UBA80 (ubiquitin 80 residue ribosomal fusion protein), encode ubiquitin fused to either ribosomal protein L40 or S27a. Furthermore, fusions of Ub have been identified in conjunction with various other genes encoding ribosomal proteins eS8, eS19, uL1, and eL41. Additionally, the gene responsible for encoding the ubiquitin-like protein FUBI (also known as MNSFβ, FUB1, or UBIM) has been found to be fused with the ribosomal protein eS30 (RPS30/FAU) [22]. This latter fusion is observed in a range of organisms, including animals and unicellular organisms, but is notably absent in yeast [23].

Typically, genes encoding polyubiquitin include four to nine head-to-tail tandem ubiquitin repeats, with the number varying depending on the organism. These ubiquitin precursors are cleaved at the Gly-Gly site at position 75–76 by specific deubiquitinases to release functional ubiquitin monomeric units. It is noteworthy that, in addition to genes encoding polyubiquitin and ubiquitin fused to ribosomal proteins, monomeric ubiquitin genes are also encoded in the genomes of a multitude of animals, Fungi, and Plants [24]. In this instance, the total level of ubiquitin is maintained primarily by the expression of the ubiquitin gene fused to the ribosomal proteins L40 and S27a. The fusion of ubiquitin with ribosomal proteins is likely responsible for enabling multicellular Eukaryotes to regulate two related cellular functions: protein synthesis and degradation. According to Martín-Villanueva et al., ubiquitin, being fused to ribosomal proteins L40 and S27a, functions as a cis-acting molecular chaperone assisting in ribosomal protein synthesis and stacking [24,25]. During embryonic development in mice, polyubiquitin gene expression has been shown to play a key role in maintaining the cellular Ub pool [26]. This suggests that the polyubiquitin gene is likely older than the ubiquitin genes that are fused with ribosomal proteins; according to Ernst Haeckel’s theory, “ontogeny recapitulates phylogeny”. Nevertheless, it is still unclear whether the primary gene is that encoding polyubiquitin or that encoding ubiquitin fused with ribosomal proteins.

Ubiquitin-like proteins of multicellular Eukaryotes include the SUMO1, SUMO2, SUMO3, NEDD8, ATG8, ATG12, URM1, UFM1, FAT10, FUBI, and ISG15 families [5]. Some members of the Ubl family, such as UFM1, ISG15, NEDD8, SUMO, and ATG8, are encoded as precursor proteins that must be cleaved by specific proteases to form the mature form. In contrast, ATG12, FAT10, and URM1 are encoded in the genome and translated in their mature form [7]. Most Ubl family precursors are known to be non-tissue-specific; however, some of them exhibit specialized functions [7]. For example, ISG15 participates in the immune response of multicellular Eukaryotes and is involved in antiviral signaling, including RIG-I, NF-κB, cytokine and chemokine production, and immune cell activation [27]. It is notable that in birds, ISG15 is absent [28], while the development of the antiviral response is mediated by an alternative Ubl, OASL (Oligoadenylate Synthetase-Like Protein), which is 37% identical to human ISG15 [29].

### 2.2. Yeast Ubls

In the genome of *S. cerevisiae*, there is only one gene encoding polyubiquitin, UBI4 (equivalent to the human genes UBB and UBC), and three genes: UBI1, UBI2 (equivalent to the human UBA52 gene), and UBI3 (equivalent to the human RPS27A/UBA80 gene). These genes encode ubiquitin fused to ribosomal proteins eL40A and eL40B (within the 60S ribosome) and ribosomal protein eS31 (within the 40S ribosome), respectively [30,31]. In actively growing yeast cells, most of the ubiquitin is produced by the expression of three genes fused to ribosomal proteins [30], which mainly provide the bulk of Ub in the cell [32]. Interestingly, the amino acid sequence of yeast ubiquitin differs from human ubiquitin by only three amino acids [32,33]. Beyond polyubiquitin, the *S. cerevisiae* genome also encodes other Ubls such as SUMO, NEDD8, URM1, ATG8, and ATG12 [34,35,36]. UFM1, however, is absent in them [37], possibly due to its important role in maintaining multicellular complexity.

### 2.3. Ubiquitin and Ubls in Protozoa

Protozoa is a polyphyletic group of organisms, which includes both parasitic and free-living forms. The genome of free-living *D. discoideum* contains one gene encoding polyubiquitin and four monoubiquitin genes, two of which are fused with ribosomal proteins [38]. In *Trypanosoma cruzi*, a parasitic protozoan, five genes have been identified that are fused with ribosomal proteins eL40 and s27a, and five genes encoding polyubiquitin have been described. The number of copies of the ubiquitin gene present within genes encoding polyubiquitin in *Trypanosoma cruzi* has been found to vary from 2 to 52 [39,40]. The authors propose that the large number of ubiquitin copies within the polyubiquitin gene can be attributed to two factors: the parasitic lifestyle of *Trypanosoma cruzi* and the need to maintain a high level of free ubiquitin. This free ubiquitin is needed for degrading host proteins, providing the parasite with more free amino acids for its metabolic needs and nutrient acquisition. Additionally, the authors cite the response to stress as another potential contributing factor.

The *Giardia lamblia* genome encodes ubiquitin fused with eS31, which is not cleaved by proteases. This phenomenon appears to be the result of a glycine substitution at position 75 for alanine in the ubiquitin structure at the junction between the ubiquitin and the eS31 protein [41]. It is probable that the original ubiquitin, situated at the N-terminus of S27a in *Giardia lamblia*, has become resistant to processing by deubiquitinating proteases due to a random mutation or structural inaccessibility. Such mutations may result in the pseudogenization of the ubiquitin gene and, potentially, the formation of a ubiquitin-like domain within the fusion protein. In the case of ubiquitin resistance to proteolysis, this freed the ubiquitin domain from negative selection, thereby allowing for the fixation of proteins containing ubiquitin-like domains in evolution.

Furthermore, the organization of the gene encoding ubiquitin in the genome of mixotrophic unicellular algae of the chlorarachniophyte group was also investigated, revealing some interesting features. In particular, the gene encoding ubiquitin can be fused with sequences encoding another ribosomal protein, P1, as well as with actin, zinc finger protein, nickel superoxide dismutase, or a protein similar to bacterial integral membrane proteins [42,43]. It is noteworthy that in the Cryptophyte *Guillardia theta* and in the Chlorarachniophyte *Bigelowiella natans*, the ubiquitin fragment can also occupy a C-terminal or even an intragenic position relative to its fusion partner [43]. The observed diversity of fusion proteins across different protozoan species is likely due to a combination of random events, including the spontaneous formation of a fused ubiquitin protein with another protein. Such fusions, termed “neutral” by Stoltzfus, can subsequently lead to the acquisition of new functions or the emergence of novel Ubls [44]. Alternatively, it may be the case that fusion proteins with ubiquitin lack any functional significance. In this scenario, ubiquitin serves merely as a “safety net” for co-expression with other genes. The question remains unanswered as to the functional significance of ubiquitin’s co-expression with proteins other than those with which it has previously been associated.

Protozoan Ubls regulate the formation and functioning of parasite-specific organelles and life forms. In particular, SUMO plays a role in the formation of the adhesive disc in *G. lamblia*. ATG8 is necessary for the maintenance of the apicoplast in apicomplexan parasites, while NEDD8 and SUMO are essential for the normal functioning of the flagellar apparatus in trypanosomes. In addition, some Ubls in parasitic Protozoa fulfill unrelated or even opposing roles to those of their homologues in other organisms. SUMOylation of chromatin has been associated with the activation of VSG transcription in *T. brucei*. However, in other organisms, this modification has been shown to repress transcription. In contrast to its usual role in ER function, UFM1 has been shown to regulate mitochondrial processes in *L. donovani* [45].

### 2.4. Archean and Bacterial Ubls

Although Prokaryotes possess their own protein degradation systems utilizing Ubls, such as Pup and Ubact, genes encoding Ubls with a structure analogous to eukaryotic ubiquitin, including SAMP1 (Archaea), CsUbiquitin (Archaea), BilA (Bacteria), and URM1 (Archaea), have been identified within the genomes of Prokaryotes. A homologue of human ubiquitin (CsUb) was identified in Archaea (*Caldiarchaeum subterraneum*), exhibiting 31% identity [46]. It is also noteworthy that the possibility of forming poly-CsUb chains is reasoned for CsUb due to the presence of Lys31, which spatially corresponds to the position of Lys29 of human Ub [47]. In Archaea, an alternative protein tagging system utilizes small modifier proteins, SAMPs, which are 18% identical to human ubiquitin [48], as well as URM1. URM1, as in Eukaryotes, has a dual function as a sulfur transporter during tRNA thiolation and protein modification [49,50] and is exclusively found in Crenarchaeota of Archaea [51].

Among α, β, γ, δ/ε—*proteobacteria*, *Actinobacteria*, and *Cyanobacteria*, operons encoding a protein with tandem repeats of the ubiquitin-like domain, polyubiquitin (PolyUbl), were identified [52]. In addition to polyubiquitin, these operons include an inactive E2-like UBC domain (the catalytic domain of the E2 conjugating enzyme), a multidomain JAB deubiquitinase fusion protein fused with the E1 domain. Bacterial polyubiquitins typically comprise two to three Ub-like domains. Some species of *α-proteobacteria* possess a distinct Ub-like domain as part of polyUbl, in addition to separate variants of polyubiquitin fused with an inactive E2-like domain [52].

In addition to polyubiquitin, operons comprising four genes encoding ubiquitination enzymes have been recently identified in bacteria [53]. These operons, specifically BilABCD, encode a homologue of the human ISG15 protein (BilA), as well as homologues of the E1 (BilD) and E2 (BilB) enzymes and DUBs of the JAB/JAMM (JAB1/MPN/MOV34 metalloenzymes) family (BilC). Bacterial BilA encodes a protein with two consecutive ubiquitin-like domains, each exhibiting structural homology to the corresponding domains of human ISG15. Prior to its enzymatic cleavage, BilA is subjected to proteolysis by the DUB BilC, which facilitates the opening of the reactive C-terminal glycine [54].

### 2.5. Ubiquitin and Ubiquitin-like Proteins: Different but Similar

We constructed a phylogenetic tree of Ubls from various Eukaryotes and Prokaryotes (Figure 1A) and found that Ubls from different groups form separate clades, regardless of whether they belong to Prokaryotes or Eukaryotes. This pattern was observed for all Ubls families. For example, the proteins of the Urm1 family of Archaea, SaUrm1 (*Sulfolobus acidocaldarius*), clustered together with Urm1 of *Dictyostelium discoideum* (DdUrm1), Urm1 of *Homo sapiens* (HsUrm1), and with Urm1 of *Saccharomyces cerevisiae* (ScUrm1) (Figure 1A). The *Dictyostelium discoideum* SUMO (DdSUMO) family proteins formed a discrete clade on the phylogenetic tree, closely related to Smt3 *Saccharomyces cerevisiae* (ScSmt3, homologue of HsSUMO1), as well as SUMO1-4 Homo Sapiens (HsSUMO1-4). The Ubls family from Hs, Dd, and Sc involved in autophagy, including ATG8 and ATG12, formed a distinct clade on the tree. A structural alignment of human ubiquitin (HsUBC), archaeal ubiquitin (CsUbiquitin), and bacterial ubiquitin-like protein (BsYukD) revealed a single three-dimensional fold (Figure 1A). The alignment of amino acid sequences of ubiquitin-like domains of proteins from different families of Ubls revealed the presence of highly conserved amino acids in the composition of Ubls in both Prokaryotes and Eukaryotes (Figure 1B).

Summarizing, it is likely that the evolutionary transformation of Ubls in Prokaryotes and Eukaryotes followed two distinct paths. On the one hand, the preservation of amino acids within Ubls was maintained to ensure the universal structure remained intact, with strict negative selection acting as the driving force. On the other hand, unique amino acid sequences were introduced by evolution to confer novel functions to Ubls.

## 3. Ubiquitin Conjugation Machinery: Billions of Years of Improvement

In accordance with a long-standing paradigm, the ubiquitin system was regarded as a distinctive feature of Eukaryotes [35]. However, mounting evidence indicates that and Archaea possess unique components of the ubiquitination system in addition to homologous components of the ubiquitin signaling system, which are structurally similar to those found in Eukaryotes. The ubiquitin signaling system of Eukaryotes is thought to have evolved by recruiting prokaryotic enzymes involved in the biosynthesis of the coenzymes thiamine and molybdopterin to perform a novel function [35,55]. In bacteria, the proteins MoaD (molybdopterin converting factor, subunit 1) and ThiS (thiamine biosynthesis protein S) exhibit structural similarity to ubiquitin [56] and function as sulfur transporters for incorporation into molybdopterin and thiazole, respectively. The activation of MoaD and ThiS is mediated by the bacterial enzymes MoeB and ThiF [57,58]. MoeB and ThiF exhibit sequence homology with the eukaryotic E1 domain, which is responsible for the binding and adenylation of Ubls and is a common building block for all E1s. Consequently, the bacterial enzymes MoeB and ThiF represent the minimal modules that ensure the recognition and adenylation of Ubls by E1 [35]. Based on these minimal modules, a variety of components of the ubiquitin and Ubls conjugation system with different substrates were generated in Eukaryotes. The emergence of multidomain proteins in early Eukaryotes represented a significant divergence from the components of the ubiquitination system of Prokaryotes, in which most genes responsible for substrate ubiquitination encode single-domain proteins [59]. The complexity of the domain organization of ubiquitination enzymes has resulted in the diversity of functions they perform in Eukaryotes.

The enzyme E1 initiates the ubiquitination process (Figure 2A). E1 activates ubiquitin via its ATP-binding domain to form the intermediate product, ubiquitin-adenylate [60]. E1 simultaneously binds both ATP and ubiquitin, catalyzing the acyladenylation of the C-terminus of the ubiquitin molecule. In a subsequent step, ubiquitin is transferred to the cysteine residue in the active center, accompanied by the formation of a thioester bond between the C-terminal carboxyl group of ubiquitin and the cysteine sulfhydryl group of E1, releasing AMP. Next, the activated ubiquitin is transferred to the catalytic UBC domain of the ubiquitin-conjugating enzyme E2. The specific binding of ubiquitin to the substrate is carried out by ubiquitin E3 ligase enzymes. E3 ubiquitin ligases consist of two distinct enzyme types. Firstly, those with a RING domain bind to E2 and the substrate, bringing them into close proximity for the transfer of ubiquitin to the substrate, a process catalyzed by the E2 enzyme. Secondly, ubiquitin ligases with a HECT domain (HECT ligases) and a TRIAD domain (RBR ligases) are directly involved in the transfer of ubiquitin to the ε-NH2 group of lysine residues of the substrate, forming an isopeptide bond. Additionally, these ligases catalyze the transfer of ubiquitin to the α-amino N-terminal group, to serine, threonine, or cysteine residues of the substrate, forming ester or thioester bonds with substrate proteins. Finally, ubiquitin moieties may be removed by deubiquitinating enzymes (DUBs), which can completely or partially detach ubiquitin chains from the substrates.

### 3.1. E1 Activating Enzyme: Snow White and the Seven Dwarfs

In addition to UBA1, seven other E1 variants are present in the human genome. E1s are classified as either canonical or non-canonical enzymes based on their structural characteristics (Figure 2B). E1 enzymes for ubiquitin (UBA1, UBA6), ISG15 (UBA7), SUMO (SAE1-SAE2), and NEDD8 (NAE1-UBA3) are categorized as canonical E1 enzymes. Canonical E1 enzymes are either mono- (UBA1, UBA6, UBA7) or heterodimeric (SAE1-UBA2, NAE1-UBA3) proteins containing several conserved domains. The adenylation domain comprises two motifs, designated “active” (AAD) and “inactive” (IAD), which are homologous to the bacterial MoeB/ThiF enzymes. The active adenylation domain (AAD) non-covalently binds Mg^2+^, ATP, and Ubl, whereas the inactive domain (IAD) provides structural stability for this interaction [56,61]. The cysteine domain is divided into two catalytic half-domains, in which the first domain is FCCH (first catalytic cysteine half-domains) and the second domain is SCCH (second catalytic cysteine half-domains). The catalytic cysteine is situated within the SCCH domain. Both half-domains are located in each of the adenylation domains. The first of these comprises a bundle of four helices, representing a second insertion into the inactive adenylation domain and forming the canonical E1 domain. The second is a C-terminal ubiquitin-fold domain (UFD) that recruits E2 [62] (Figure 2B).

Ubiquitin proteins bind to Mg^2+^-ATP in the active adenylation domain where the adenylation of the C-terminal glycine occurs, resulting in the formation of Ub-adenylate [Ub-adenosine-5′-monophosphate (Ub-AMP)] and pyrophosphate (PPi). The release of PPi triggers conformational changes that allow the nucleophilic attack of the acyladenylate by active cysteine residue [63]. This forms a thioester bond and likely pulls ubiquitin away from the AAD, while the reactive cysteine of the SCCH domain returns to its original position upon AMP release. Subsequently, the active adenylation site can bind a second Ub molecule, creating an E1 double Ub complex. In this complex, one Ub is covalently bound to the catalytic cysteine domain (SCCH) (Uba1~Ub) and the second Ub is noncovalently bound in the active adenylation site (Figure 2C). The interaction of E1 with E2 via the C-terminal domain results in the formation of an E1-E2-Ub_2_ ternary complex via a ubiquitin-fold domain that recruits E2 enzymes and positions the catalytic cysteines of E1 and E2 in close proximity. This facilitates the transfer of ubiquitin from E1 to E2 [54,64]. Structural studies of ubiquitin and SUMO E1 ligases have revealed that the SCCH domain undergoes a 130° rotation upon adenylation of Ub/SUMO [65]. In 2019, research further detailed these conformation changes, showing remodeling of the α-helix containing the catalytic cysteine, as well as the N-terminal helices of the IAD domain, which coordinate ATP and pyrophosphate. Additionally, the cross- and re-entry loops for Ubl underwent rearrangement [62,65]. Similar structural transitions were observed in Uba1 from *S. pombe*, where the SCCH domain rotated by 106°. Considering the fact that SUMO and Ub E1 represent divergent members of the canonical Ub/Ubl E1, Hann et al. hypothesized that similar domain alternations would underlie adenylation cycles and thioester bond formation in other canonical E1s as well [65].

The E1 enzymes for ATG12 and ATG8 (ATG7), for URM1 (MOCS3/UBA4), and for UFM1 (UBA5) are classified as non-canonical (Figure 2B). Like other bacterial MoeB and ThiF, non-canonical E1s have a symmetric structure and can form homodimers. In contrast to canonical E1 enzymes, non-canonical E1 enzymes lack a distinct catalytic Cys domain. Instead, the catalytic cysteine residue is embedded within the adenylation domain. In addition to the AAD, non-canonical E1s possess unique sequences (Figure 2B).

In addition to the adenylation domain, ATG7 contains binding sites for E2 (ATG3, ATG10) and ubiquitin-like proteins (ATG8, ATG12). ATG7 binds ATG3 and ATG10 (E2) via an N-terminal domain, and ATG8 and ATG12 (Ubls) via an adenylation domain and a C-terminal IDR (intrinsically disordered region). Structural studies of ATG7 have revealed a multistep pattern of ATG8 recognition by ATG7. Initially, ATG7 interacts with ATG8 via the C-terminal IDR, transferring ATG8 to the adenylation domain. The C-terminal tail of ATG8 is then placed in the adenylation site, where it undergoes thioester formation with the catalytic Cys of ATG7 and the C-terminal Gly of ATG8 [66].

Another non-canonical E1, MOCS3/Uba4 (E1 for URM1), is notable for its structural similarity to numerous bacterial MoeB and ThiF family proteins. These include the presence of a C-terminal rhodanase domain (RHD), which comprises approximately 120 amino acid residues (Figure 2B). Rhodanases and a number of RHD proteins are sulfur transferases that transport sulfur to their targets via a persulfide (-S-S-H) intermediate on the cysteine of their active center. Furthermore, Uba4 and MOCS3 are involved in the uridine thiolation pathways of selected tRNAs (tRNALys with the anticodon sequence UUU, tRNAGln(UUG), and tRNAGlu(UUC)), where they act as sulfur donors [67,68]. The presence of RHD allows Uba4 and MOCS3 to conjugate Urm1 to target proteins in an E2/E3-independent manner via a cysteine persulfidation reaction [69]. It is, therefore, probable that the first Eukaryotes were able to transfer Urm1 without the involvement of E2 and E3 through the adenylation and rhodanase domains. The E1 enzymes vary across different groups of organisms, as does that of Ubls.

### 3.2. Eukaryotic E1

A total of eight E1 ubiquitin-activating enzymes have been identified in humans: for ubiquitin (UBA1, UBA6), for FAT10 (UBA6), for SUMO 1-3 (SAE1-SAE2 (UBA2)), for NEDD8 (NAE1-UBA3), and for ISG15 (UBA7), as well as E1s for URM1 (UBA4), UFM1 (UBA5), ATG12, and ATG8 isoforms (ATG7). It is noteworthy that UBA1 and UBA6 exhibit approximately 40% sequence identity, with UBA1 being phylogenetically closer to UBA7 (E1 for ISG15). UBA6 and its specific E2 enzyme, USE1, are ubiquitously expressed in humans, *Danio rerio*, and sea urchins but are absent in worms, flies, and yeast. This suggests that USE1 plays a selective role in individual multicellular organisms [70]. Furthermore, USE1 is an E2 enzyme that not only acts on ubiquitin but also on FAT10 [71], which determines the cross-reactivity of E2 against different ubiquitin-like proteins (Ubls). In yeast, six genes encoding E1 enzymes have been identified (UBA1, ATG7, UBA3, UBA4, UblE1A-UbLE1B, and UBA2, which encodes the SMT3 protein, an analogue of the mammalian SUMO1 protein). In particular, the protozoan *Dictyostelium discoideum* has been found to possess 12 genes encoding E1 enzymes, 2 of which are responsible for the synthesis of ubiquitin, while the remaining 10 are involved in the synthesis of SUMO, NEDD8, ATG8, ATG12, URM, FAT10, and ISG15 proteins. Notably, the genome of *Dictyostelium discoideum* encodes E1 orthologues, ubiquitin-conjugating enzymes for FAT10 and ISG15 proteins, in the absence of genes encoding these Ubls [38].

### 3.3. Prokaryotic E1 Analogues

A long-standing paradigm suggests that eukaryotic E1 enzymes originated from prokaryotic components of biosynthetic pathways. MoeB and ThiF, which exhibit sequence homology with the adenylation domain of eukaryotic E1 enzymes, represent minimal modules with the activity of recognizing and adenylating Ubls and forming a thioester bond with them [35]. In addition to MoeB and ThiF, operons encoding the main ubiquitination enzymes have been recently identified in bacteria, most frequently among *α-proteobacteria* [53]. The BilABCD operon, for instance, encodes a homologue of the human ISG15 protein (BilA); the Bil operon includes genes for proteins homologous to E1 enzymes (BilD) [54]. E1 BilD comprises an N-terminal inactive adenylation domain (IAD) and a C-terminal active adenylation domain (AAD), which together form a structurally related pseudodimer. The AAD of E1 BilD also includes an approximately 80-amino-acid α-helical domain with a conserved cysteine residue (Cys domain), resembling the catalytic Cys domain of eukaryotic enzymes. The closest structural relatives of E1 BilD are the human adenylation domain E1 UBA6 (E1 for ubiquitin/FAT10) and the heterodimeric human E1 NAE1-UBA3 (E1 for NEDD8). E1 BilD activates BilA in an ATP-dependent manner, and E1 BilD (Cys417) reacts with BilA adenylate to form the intermediate E1 BilD~BilA [54].

The genomes of some Archaea (Euryarchaea) contain genes encoding MoeB, which are involved in the biosynthesis of the tungsten cofactor. Additionally, these archaeal genomes encode E1 activating enzymes specific to ubiquitin-like proteins SAMPs and Urm1 [72]. Interestingly, they lack the catalytic Cys domain homologous to that found in Eukaryotes. However, Archaea from the superphylum Asgard (*Caldiarchaeum subterraneum*) were shown to possess a gene cluster that, beyond Ubls, E2 enzymes, DUB, and RING E3 ligase, encompasses E1 enzymes. These E1 enzymes have a distinct domain (60 amino acids) housing a catalytic cysteine [47]. Three-dimensional models of the E1-like enzyme of *C. subterraneum* reveal the preservation of key structural features and catalytic residues characteristic of the ubiquitin-activating E1 domain in Eukaryotes, particularly mirroring those found in the NEDD8-activating E1 enzyme, Uba3 [47]. Evolutionary analyses based on bacterial and archaeal genomes indicate that the LUCA already possessed proteins containing an E1-like domain, potentially including a separate catalytic Cys domain. Furthermore, these E1-like proteins were not only involved in thiolation activity and multiple sulfoprotein transfer reactions, but also played a role in the activation of ubiquitin-like proteins [52].

### 3.4. Evolutional Profiling of E1 Activating Enzyme

It is hypothesized that proteins evolve not only through point mutations but also via modular rearrangements at the level of protein domains, which are considered the units of modular evolution [73]. In our review, we aimed to identify homologues of human E1 catalytic domains among Eukaryotes, Bacteria and Archaea, based on currently annotated genomes. To accomplish this, we performed a protein–protein Blast analysis (v. 2.12.0) of the amino acid sequences of human E1 catalytic domains against all annotated sequences in the NCBI database. Analysis of catalytic domains but not full sequences was reasoned by a high degree of false-positive homology that occurred due to the significantly increased frequency of other domains, e.g., the adenylation domain, in proteins completely not related to ubiquitin ligases. Our global evolutionary analysis of E1 ligase homology across all currently living organisms uncovered several notable findings (Figure 3, Appendix A). Among Archaea and Bacteria, we identified homologues of the catalytic domains of the non-canonical E1 catalytic domains of human MOCS3 and UBA5 (Figure 3A). Within these groups, the total score values for the catalytic domain of MOCS3 were approximately three times higher than those for UBA5, indicating that E1 UBA5 may be a more distant homologue for the catalytic domain of human UBA5 than MOCS3 is. This finding suggests that UBA5 could be considered more ancient.

No homologues of the catalytic domain of the non-canonical human E1 enzyme ATG7 were identified in Archaea and Bacteria (Figure 3A). This absence is likely attributable to the lack of membrane-bound organelles in Prokaryotes and their distinct mechanisms for intracellular proteolysis of misfolded proteins, which involve enzymes located in the periplasmic space of Gram-negative bacteria, the extracellular matrix of Gram-positive bacteria, and the quasi-periplasmic space in Archaea [74,75]. Furthermore, it is hypothesized that ATG conjugation systems evolved independently from ubiquitination systems [76], suggesting that the ATG system is a unique eukaryotic acquisition. Additionally, identifying the most probable common ancestor of the catalytic cysteine (Cys) domains UBA1, UBA2, and UBA3 in Bacteria and Archaea has proven challenging. This difficulty is probably due to significant amino acid changes that have occurred throughout their evolution. As a result, the question regarding the origin of the catalytic Cys domain of these E1 ligases in Eukaryotes remains unresolved.

The analysis of the catalytic domains of MOCS3 and UBA5 within the Bacteria kingdom revealed considerable variation in total score values (Figure 3B, Appendix A). The highest total score values for bacterial MOCS3 were found in unculturable bacteria from the following phyla: *Candidatus Kryptonia*, *Candidatus Abyssubacteria*, *Candidatus Handelsmanbacteria*, *Candidatus Abyssubacteria*, *Acidobacteriota*, *Armatimonadota*, *Chloroflexota*, *Deinoccocota*, and *Myxococcota*. Conversely, the lowest score values for MOCS3 in bacteria were observed in *Myxoplasmatota* and *Chlamydia* phyla, possibly due to the parasitic lifestyles of many species of these phyla. For UBA5, the highest values were observed in species *Candidatus Kryptonia* and *Candidatus Lithacetigenota*. Within the domain of Archaea, the highest score values for MOCS3 and UBA5 were observed in *Candidatus Geothermarchaeota*, *Candidatus Helarchaeota*, *Candidatus Hadarchaeota*, *Halobacteriota*, and *Candidatus Nezhaarchaeota*.

Among the E1 enzyme domains, those containing a catalytic cysteine (Cys) residue—specifically, UBA1, UBA6, UBA7, UBA3, and UBA2—as well as the FCCH domains of NEA1 and SAE1 were identified as unique to Eukaryotes, as anticipated (Figure 3A). Interestingly, the FCCH domain of SAE1 was found to be specific to the chordate lineage (Figure 3B and Appendix A). The human homologue of SAE1—AOS1—is also found in Fungi (*Saccharomyces cerevisiae*), Plants (*Arabidopsis thaliana*), and Protozoa (*Plasmodium falciparum*). AOS1, like human SAE1, forms a heterodimer with UBA2 to activate SUMO-like proteins. A multiple sequence alignment of the full-length AOS1 and SAE1 proteins revealed matches in several amino acids located outside the FCCH domain of human SAE1, suggesting a potentially unique structural role of the FCCH domain in chordates compared to other Eukaryotes. It is likely that the formation of single subunits, which later formed heterodimers with UBA3-NEA1 and UBA2-SAE1, preceded the emergence of multidomain monomers, such as UBA1, UBA6, and UBA7.

According to Burroughs et al., UBA1, UBA2, and UBA3 may have originated with the LECA. UBA6 is thought to have emerged from the common ancestor of Fungi and multicellular Eukaryotes, while the UBA7 gene is believed to have arisen from the duplication of the UBA1 gene in vertebrates [60]. Our findings indicate that the catalytic domains of UBA3, UBA1, and ATG7 (except for the Rhodophyta, in which autophagy genes have been lost [77]) are among the most conserved E1 enzymes across all Eukaryotes (Figure 3A,B). In addition to its role in autophagy, ATG7 participates in various cellular processes, including responses to hyperoxia, starvation, protein transport and lipidation, circadian rhythm regulation, the positive regulation of the promotion of apoptosis, defense against viruses, and the regulation of the cell cycle. UBA1 plays an important role in the DNA damage response, whereas UBA3 is involved in proteolysis, protein modification, and the regulation of the cell cycle through post-translational modifications (Figure 4, Appendix A).

Our analysis reveals that, in addition to the previously mentioned E1 enzymes (UBA3, UBA1, ATG7), the catalytic domains of UBA2 and NAE1 are among the most conserved in the Metazoa (Figure 3B and Appendix A). NAE1 plays an important role in mitotic DNA replication checkpoint signaling, signal transduction, and the regulation of apoptosis. The observed high degree of conservation in the catalytic domains of UBA2 and NAE1 within metazoan species likely reflects the essential need to maintain multicellularity. This preservation is facilitated through distinct cellular functions mediated by SUMOylation and NEDDylation (Figure 4). In contrast, the catalytic domains of MOCS3, UBA5, UBA6, and UBA7 were less affected by evolutional selection, which allowed for the diversification of regulatory intracellular signaling networks involving the ubiquitination system in Metazoa.

Among Eukaryotes, the most variable E1 domains of MOCS3, UBA5, UBA6, and UBA7 exhibit the highest variability (Figure 3B and Appendix A). It has been postulated that MOCS3 and ATG7 were among the first E1 enzymes to emerge, predating the formation of the LECA [60]. Contrary to this hypothesis, our data indicate that ATG7 is the most conserved and uniquely present in Eukaryotes, while MOCS3 shows the greatest variability. It is noteworthy that UBA6 plays an important role in nervous system development, influencing the formation of key brain structures such as the amygdala and hippocampus, as well as dendritic spine development, locomotor behavior, and learning processes (Figure 4). Knockout studies of neuronal UBA6 during embryonic development have demonstrated altered neuronal patterning in the hippocampus and amygdala, reduced dendritic spine density, and various behavioral disorders [78]. The diversity observed in the UBA6 catalytic domain likely reflects evolutionary adaptations related to the formation and functionality of the hippocampus and amygdala across different organisms, as well as differences in their neuronal plasticity and behavior.

The degree of E1 variation in different types of Metazoa is represented in disparate ways. The median values for the total score, presented as boxplots (Figure 3C), demonstrate that Echinodermata, Mollusca, Arthropoda, Annelida, and Cnidaria have median values for the total score similar to those for Chordata E1, probably as a result of the convergent evolution of E1 enzymes. In contrast, Nematoda, Rotifera, and Platyhelminthes are characterized by slightly lower median values compared to Chordata E1. The lowest total score values for homologues of the human UBA1 catalytic domain are observed in *Protista* species belonging to Ciliophora and Euglenozoa, as well as Apicomplexa, many of which are parasitic. Phyla with less than six individual annotated genomes were withdrawn from analysis.

### 3.5. Who Is Next? The E2 Conjugating Enzyme Superfamily

Following the activation of E1, ubiquitin is transferred to the E2 conjugating enzyme. All E2 ubiquitin-conjugating enzymes share a common central catalytic domain, the UBC domain (140–200 amino acids), which contains a cysteine residue at the active center (Figure 2). The UBC domain is primarily comprised of four α-helices and an antiparallel β-sheet consisting of four β-bands. E2 proteins are highly homologous to one another. E1 interacts with α-helix 1, while E3 interacts with loop 1 and loop 2 of the β-bands [79]. The E2 catalytic domain of Eukaryotes also includes a highly conserved HPN (His-Pro-Asn) amino acid sequence upstream of the Cys residue used for ubiquitin conjugation. It has been postulated that the asparagine residue stabilizes an oxyanionic intermediate formed in the thioester E2~Ub/Ubl during the nucleophilic attack of the lysine residue of the substrate [80,81]. The amide group of the asparagine side chain interacts with the thioester carbonyl, stabilizing the oxyanion of the intermediate and maintaining the correct orientation of the carbonyl carbon atom for a nucleophilic attack by the lysine residue of the substrate. This stabilization of the oxyanion intermediate also facilitates the transfer of Ub/Ubl from E2 to the cysteine residue of the active site cysteine of HECT/RBR E3 ligases to form the thioester E3~Ub/Ubl. In this instance, histidine serves a structural role, forming a hydrogen bond with asparagine within the HPN, without directly influencing the catalysis of Ub transfer [82]. Consequently, the highly stable HPN structure depends on the precise orientation of histidine, which correctly positions the HPN asparagine to interact with the Ub/Ubl intermediate thioester, thereby facilitating the attachment of Ub/Ubl to the substrate [82].

E2 enzymes are classified into four categories based on the presence of extensions at the N- and/or C-termini (Figure 2). Class I comprises enzymes that contain only the ubiquitin-binding domain (UBC). Class II includes enzymes with the UBC domain in addition to an N-extension. Class III contains enzymes with the UBC domain in addition to a C-extension. Finally, class IV encompasses enzymes with both N- and C-extensions in addition to the UBC domain [79,83]. These extensions are often intrinsically disordered, though some acquire a secondary structure that binds to the UBC domain [84]. Functionally, these additional domains determine the intracellular localization of E2 enzymes, serve regulatory functions, and provide specific interactions with distinct E3 enzymes [85]. Furthermore, some E2 proteins are part of larger multidomain proteins, including Ube2O, UBE2Z, and BIRC6.

The E2 conjugating enzymes vary across different groups of organisms. In humans, 40 E2 enzymes have been identified. While the majority of E2s are ubiquitously expressed, some exhibit preferential expression in specific tissues. For instance, UBE2U is expressed in the urogenital tract, UBE2O is found mainly in skeletal and cardiac muscles, and high levels of UBE2K expression have been identified in specific regions of the human brain [86]. This suggests that the transition to multicellularity and the consequent need for tissue-specific regulation contributed to the diversity of E2 enzymes observed in the Metazoa kingdom.

In *S. cerevisiae*, 14 E2 enzymes have been identified. Apicomplexa species, such as *Plasmodium* spp. and *Toxoplasma gondii*, have approximately 8–14 genes encoding E2 enzymes [87]. In *P. falciparum*, nine out of fourteen E2 enzymes exhibit stage-specific expression [87], likely reflecting the complexity of its life cycle. In *D. discoideum*, approximately 30 genes encoding E2 enzymes have been identified. The number of E2 enzymes in *D. discoideum*, which is comparable to that in Apicomplexa species and is similar to that in humans, is probably associated with its unique morphogenesis and potential transition to a multicellular form.

Homologues of eukaryotic E2 enzymes have also been identified in Prokaryotes. In Archaea, in contrast to Eukaryotes, E2 and E3 enzymes are not required for the conjugation of small archaeal modifier proteins (SAM) to proteins (Humbard et al., 2010). However, in Archaea from the superphylum Asgard (*Caldiarchaeum subterraneum*), E2 enzymes structurally homologous to eukaryotic E2 enzymes have been discovered [6,47]. The structural similarity between eukaryotic and archaeal E2 enzymes extends to the conservation of key amino acid residues that mediate interactions with E3 ligases [47].

The bacterial E2 enzyme BilB, from the operon BilABCD, contains a catalytic cysteine residue (Cys138) as part of the conserved CEHH motif (Cys138, Glu144, His146, and His151). Cys138 is directly involved in catalysis, as is His151, which is adjacent to Cys138. Two additional conserved residues, Glu144 and His146, appear to play a structural role in E2 BilB, with their side chains engaged in hydrogen bonding interactions with nearby amide and carbonyl groups. The closest structural homologues for E2 BilB are UBC12 of *S. cerevisiae* and UBE2D2, UBE2D3, and UBE2J2 of *H. sapiens* [54]. The presence of histidine in bacterial E2 enzymes is analogous to that observed in the His-Pro-Asn E2 sequence found in Eukaryotes. This suggests the existence of an ancient mechanism for positioning key amino acids within E2 enzymes to interact with the Ub/Ubl thioester and form a Ub/Ubl conjugate with the substrate. While the exact origin of E2 enzymes remains unclear, it is known that they function outside the canonical Ub E1/E2/E3 transport pathways [74,88]. This implies that E2 enzymes may have originated from orthologues of other cascades through a process of subfunctionalization.

### 3.6. Evolutional Profiling of E2 Conjugating Enzymes

As an amino acid sequence of E2 ligases consists mainly of the UBC domain with short N- or C-terminal extensions (Figure 2B), we examined the homology of the catalytic UBC domains of E2 conjugating enzymes (generally representing almost full protein sequences) across Eukaryotes and Prokaryotes. To achieve this, we divided the UBC domains of 40 human E2 ligases into 31 clusters based on the degree of amino acid sequence identity (Figure 5A, Appendix A). The consensus sequences of these clusters were used for alignment; the total score values are given in Appendix A. According to Iyer et al., bacterial members of the E2 superfamily are encoded as operons together with E1-like, Ub-like, and Jab-peptidase components of the ubiquitination system [52]. Additionally, putative bacterial E2 families exhibit histidine and asparagine residues within their active centers, analogous to the classical E2 UBC domain of Eukaryotes [52]. Burroughs et al. proposed that the ancestral version of E2 in Eukaryotes originated from this family of proteins [76]. However, our search for homologues of the catalytic domains of human E2 enzymes in bacteria revealed only single instances in the phyla *Nitrospinota*, *Candidatus Riflebacteria*, *Vulcanimicrobiota*, and *Candidatus Aminicenantes*.

Among Eukaryotes, the most conserved E2 catalytic domains were ATG3, UBE2V1-V2, UBE2H, and PEDS_UBE2V1 (Figure 5B and Appendix A, group 5). The functions of this E2 group include macroautophagy, cell differentiation, post-replication repair, negative regulation of phagocytosis, positive regulation of intracellular signal transduction pathways, positive regulation of canonical NF-kB signal transduction and intracellular signal transduction, and regulation of the cell cycle and cell growth (Figure 6). Consequently, the emergence of autophagy and cell differentiation—functions specific to Eukaryotes—is associated with the participation of the ubiquitination system components in these processes.

The catalytic domains UBE2Q1-Q2 and UBE2QL1 are highly conserved and specific to the Metazoa kingdom, with the exception of Acanthocephala. This phylum consists entirely of obligate parasites of aquatic and terrestrial vertebrates, which may explain the absence of these domains. The total scores for UBE2Q1-Q2 and UBE2QL1 in Fungi and Protista kingdoms were found to be lower than in Metazoa. Among Fungi, UBE2Q1-Q2 and UBE2QL1 were primarily detected in Ascomycota and Basidiomycota, while they were absent in Apicomplexa, Foraminifera, Fornicata, Oomycota, and Perkinsozoa among Protista. This suggests that the catalytic domains UBE2Q1-Q2 and UBE2QL1 are relatively recent additions to the E2 family, likely emerging during the evolution of Ascomycota and Basidiomycota. The appearance of UBE2Q1-Q2 and UBE2QL1 is functionally associated with the expansion of ubiquitination substrates. This is evidenced by the E2 family UBE2Q, which ubiquitinates not only serine and threonine protein residues, but also glucose and complex sugars. The UBE2Q family differs from canonical E2s in several aspects. A notable difference is the absence of the conserved histidine–proline–asparagine (HPN) triad that characterizes canonical E2s. The non-canonical activity of UBE2Q1 depends on the presence of Tyr343, His409, and Trp414 [90]. It is also noteworthy that the highest expression level of UBE2Q1 is observed in the brain. Furthermore, UBE2Q1 has been demonstrated to exhibit pleiotropic effects during implantation and embryo development [91]. Additionally, it is involved in the development of the reproductive system, mating and breast behavior, and prolactin secretion. In addition to UBE2Q1-Q2 and UBE2QL1, metazoan-specific E2 includes TSG101 and UEVLD, which are involved in regulating MAP kinase activity, extracellular exosome assembly, the cell cycle and cell growth, membrane division, and positive regulation of ubiquitin-dependent endocytosis.

Among Metazoa, the catalytic domain UBE2C exhibited the greatest degree of variability. In Nematoda and Acanthocephala, the UBE2C gene is absent. In Protista, UBE2C homologues were absent in almost all species, with the exception of Heteroloboseans, Haptophytes, Evoseans, and Euglenozoans. UBE2C plays a pivotal role in the regulation of anaphase (APC/C) during mitosis, acting as an essential component of the complex/cyclosome. By initiating Lys-11-related polyubiquitin chains on APC/C substrates, UBE2C induces the degradation of these substrates by the proteasome and promotes exit from mitosis, thereby regulating the transition from metaphase to anaphase. Furthermore, during the process of neurogenesis, Ube2c expression is strongly repressed during the development of the rat embryo [92]. Interestingly, the median values for E2 do not differ significantly between metazoan types (Figure 5C). Phyla with less than six individual annotated genomes were withdrawn from analysis.

Our data indicate that the ancestors of the UBC domain of E2 in Eukaryotes are most likely archaeal species. In Archaea, we identified a “core set” of E2 proteins (Figure 5B and Appendix A, group 1), which encompasses 14 E2 clusters. The same E2 set is represented in all Eukaryotes, with comparable total score values among themselves except for Apicomplexa, for which homologues are shown only for UBE2D1-4, UBE2N, UBE2L5-6, UBE2R-CDC34, and with Archaea. Archaeal E2 recruitment is involved in the regulation of a number of fundamental cellular processes, including proliferation, DNA damage response, cell cycle phase transition, apoptosis, DNA metabolism, chromatin organization and remodeling, chromosome segregation, protein metabolism, and stabilization. In addition, it is involved in the negative regulation of signaling pathways, including mTOR signaling, cAMP-mediated signaling, the BMP signaling pathway, and the positive regulation of the Wnt signaling pathway and canonical NF-kappaB signal transduction. Furthermore, archaeal E2 recruitment is involved in the response to abiotic stimuli, stress, and in the antiviral innate immune response. Given the presence of this E2 set in Archaea and all eukaryotic groups, their performance of basic cellular functions is essential for maintaining the integrity and development of both single-celled and multicellular organisms (Figure 7, Appendix A).

The use of multidomain proteins in early Eukaryotes has been found to represent a key difference from archaeal systems, in which all genes encode single-domain proteins [59]. Our data indicate that individual Archaea belonging to *Candidatus Borrarchaeota*, *Candidatus Helarchaeota*, and *Candidatus Sigynarchaeota* also possess homologues for human multidomain E2, BIRC6, UBE2O, UBE2Z, and UBE2H. This suggests that these groups underwent domain swapping via horizontal gene transfer (HGT) to form multidomain E2 enzymes and probably gave rise to eukaryotic multidomain E2 proteins.

In Eukaryotes, in addition to the “archaeal set” of E2, there is also a group of E2 enzymes including UBE2J1, UBE2J2, BIRC6, UBE2O, UBE2W, and UBE2Z, for which the total score values are approximately 30 (Figure 5, group 3). The increasing complexity of the domain composition of E2 enzymes in Eukaryotes is likely due to the increasing complexity of signaling pathways [59]. The exception is Apicomplexa, for which homologues were identified only for UBE2J1, UBE2J2, UBE2W, and UBE2Z. The Apicomplexa group primarily comprises parasitic forms, which may explain why a limited set of E2 catalytic domains is encoded in their genomes. In terms of function, this group of E2 enzymes, in addition to the basic set of cellular functions (e.g., cell cycle, cell division, cell proliferation), is involved in retrograde transport from endosomes to the Golgi apparatus and its regulation, the ERAD pathway, regulation of cytokinesis, regulation of cell communication, intracellular transport and programmed cell death, cell cycle and proliferation, and response to chemical stimuli and stress (Figure 8, Appendix A). It is likely that the presence of additional domains in the E2 structure in some Archaea and most eukaryotic species, due to the expansion of the spectrum of protein–protein interactions with targets, has increased their functional repertoire, mainly within the regulatory system.

### 3.7. The Army of Conjugators: E3 Ligases

E3 ligases provide the transfer of ubiquitin from E2 ligases to the substrate. Three distinct families of proteins are identified among E3 ligases: the RING family of E3 ligases, including U-box E3 ligases, the HECT family, and the RBR family (RING-between-RING) E3 ligases. A novel family of E3 ligases, designated PCAF_N, in addition to E3 ligase activity, exhibits acetyltransferase activity and is specific to the Metazoa kingdom [93]. E3 ligases of the RING family facilitate the transfer of ubiquitin from E2~ubiquitin to the substrate through a non-covalent interaction with ubiquitin. The canonical RING domain is composed of seven cysteines and a histidine of Cys-X2-Cys-X(9-39)-Cys-X(1-3)-HisX(2-3)-Cys-X2-Cys-X(4-48)-Cys-X2-Cys motif. The cysteine and histidine residues within the RING domain serve to maintain its structural integrity by binding to two zinc atoms. In the U-box domain of E3 ligases, the zinc-binding sites are replaced by conserved charged and polar residues. PCAF_N ligases possess three conserved cysteines within their catalytic domain, which form part of the hydrophobic core. These cysteines are unable to covalently bind ubiquitin, in a manner similar to that observed in RING ligases [93].

Among RING E3 ligases, several families are distinguished: monomeric, homodimeric, heterodimeric, and multisubunit, including cullin-RING ligases, APC/C ligases, and other E3 ligases (Figure 2B). SCF E3 ligases represent one of the largest cullin-RING E3 ligase families, comprising a complex of Skp1, Cullin1, and F-box proteins. The F-box protein is responsible for the specific targeting of phosphorylated substrates for degradation. The Skp1 protein is necessary for binding the catalytic core of the SCF complex to the F-box protein. Cullin1 is essential for regulating the binding of other components of the SCF complex [94]. In mammals, six canonical cullins (Cullin1, Cullin2, Cullin3, Cullin4a, Cullin4b, Cullin5) and three atypical cullins (APC2, Cullin7, and PARC) have been identified, which together form more than 500 different multisubunit complexes within cullin-RING E3 ligases [95]. APC/C (anaphase-promoting complex/cyclosome) E3 ligases are the principal ubiquitin ligases required for the progression of mitosis and meiosis. The APC/C targets cell cycle-related proteins, such as cyclins and securin, thus ensuring the timely degradation of key cell cycle regulators [96].

HECT and RBR E3 ligases are responsible for attaching ubiquitin to the substrate via a two-step reaction (Figure 2B). The initial step is a trans-thioesterification reaction, during which the E3 enzyme accepts ubiquitin from the ubiquitin-loaded E2 enzyme and forms a thioester bond between its catalytic cysteine residue and the C-terminus of ubiquitin. In the second step, the activated C-terminus of ubiquitin undergoes a nucleophilic attack by the primary amino group of the target protein, resulting in the formation of an isopeptide bond between ubiquitin and the target protein. The HECT E3 ligase consists of an N-terminal lobe (N-lobe) containing the E2 binding domain and a C-terminal lobe (C-lobe) which forms a HECT domain with the catalytic cysteine. The two lobes are connected by a flexible hinge region, which allows the C-lobe to move, thus facilitating the transfer of Ub from E2 to E3 [97,98,99]. Additionally, other domains situated upstream of the HECT domain facilitate the specific attachment of substrates for ubiquitination [100]. The human HECT E3 ligase family consists of three subfamilies: HERC, NEDD4, and other HECT ligases [101].

RBR-type E3 ligases comprise two RING domains (RING1 and RING2) separated by an IBR (In Between RING) domain (Figure 2B). The RING1 domain is analogous to the original RING, whereas RING2 is not a real RING domain and possesses a single catalytic cysteine residue that enables it to accept a ubiquitin molecule from the E2 enzyme, form a thioester bond with ubiquitin, and transfer it to the substrate. It is also noteworthy that the IBR domain exhibits the same structural fold as the RING2 domain but lacks the catalytic cysteine residue and ubiquitination activity. The activity of numerous RBRs is contingent upon the catalytic triad of cysteine, histidine, and glutamic acid or glutamine. The histidine residue, located two residues away from the catalytic cysteine of RBR ligases, promotes deprotonation of the acceptor lysine [102]. Similarly, HECT E3 enzymes also contain a conserved histidine residue two residues away from the catalytic cysteine, which also affects the chemical environment of the catalytic center [99,100].

The E3 ligases exhibit the greatest diversity, due to their tissue- and stage-specific expression [103] and their representation in different groups of organisms. For example, approximately 700 E3 ligase genes are encoded in the human genome, 600 of which are related to the RING family; 28 belong to the HECT family and 14 to the RBR family [103,104]. Approximately 60 to 100 E3 ligases have been identified in *S. cerevisiae*, with the majority belonging to the RING E3 type [44], five to the E3 type with HECT domains, and two to the RBR E3 type [32,70]. In the genome of *P. falciparum*, only 50 E3 ligase genes have been annotated [105]. Concurrently, approximately 150 distinct E3 ligases are encoded within the genome of the slime mold *D. discoideum*, of which six are HECT-type and eight are RBR-type. *Dictyostelium discoideum* typically exists as solitary soil amoebae, but under certain conditions, it forms mobile aggregates and then multicellular fruiting bodies of complex structure. These fruiting bodies are involved in intercellular signaling, cell differentiation, and a complicated morphogenesis. Thus, such diversity in *D. discoideum* is likely to be related to the peculiarities of its morphogenesis and the possibility of transition to a multicellular form.

### 3.8. Prokaryotic E3 Analogues

As reported by Pisano et al., homologues of eukaryotic E3 ligase enzymes have also been identified in Prokaryotes [106]. Similar to eukaryotic E3 ligases, bacterial analogous are RING/U-Box ligases, HECT ligases, and also NEL-type E3 ligases. However, bacterial RING ligases diverge from canonical eukaryotic E3 ligases in the arrangement of coordinating residues within the RING domain. In contrast to the canonical C3HC4 arrangement observed in Eukaryotes, prokaryotic RING domains exhibit a C4HC3 configuration. Another feature of prokaryotic RING domains is the presence of a longer insertion between the last pair of coordinating residues. Additionally, the RING domains identified in bacteria lack two pairs of Zn-chelating residues but retain two pairs of residues from the central hairpin and the C-terminal extension [106]. The sequences and structures of bacterial HECT ligases are strikingly divergent from those of eukaryotic ligases, yet they exhibit a comparable degree of coordination. Two homologous proteins, SopA from *Salmonella* sp. and NleL from enterohemorrhagic *Escherichia coli*, have been demonstrated to exhibit E3-type HECT ligase activity [107]. Both SopA and NleL contain a cysteine residue near the C-terminus that forms a transient thioester bond with Ub [107]. Similar to eukaryotic HECT E3, SopA and NleL function with an E2 subunit containing a conserved phenylalanine residue. The crystal structures of SopA and NleL exhibit a common fold, comprising three domains: a β-helical domain that may serve as a substrate binding site; a central elongated domain, which resembles the N-lobe eukaryotic HECT domain; and a C-terminal globular domain, which resembles the C-lobe HECT E3s [108]. The trans-thioesterification reaction in bacteria occurs spontaneously when the two active cysteines are in close proximity, as observed in the UbcH7/NleL structure [90,109]. Similar to E3 ligases of the HECT family, bacterial NEL ligases contain a conserved cysteine residue that is located in a conserved CXD motif and acts as a catalytic nucleophile in the ubiquitination reaction. In addition to the catalytic cysteine, the bacterial E3 ligase IpaH9.8 also relies on the conserved residues Asp339 and Asp397, which are located near the catalytic center. These charged residues are thought to maintain a favorable electrostatic environment for catalysis. Furthermore, the Asp365 residue in the active CXD NEL center of the IpaH3 ligase (IpaHc/IpaH1383) is important for its enzymatic activity [110].

In Archaea, an E3 ligase, which belongs to the RING-H2 class of proteins defined by the sequence of zinc-binding residues C3H2C3, is also encoded in the *C. subterraneum* genome as part of the operon encoding the major components of the CsUb, E1, and E2 DUB ubiquitination cascade [47]. The Archaeon *Candidatus Prometheoarchaeum syntrophicum*, the most closely related cultured archaeal organism to Eukaryotes, is believed to possess the ubiquitin-to-substrate conjugation system most similar to that observed in Eukaryotes [111].

### 3.9. Evolutional Profiling of E3 Ligases

Given the considerable diversity of E3 functions, we performed a search for homologues of human E3 domains among Eukaryotes, Archaea, and Bacteria. To achieve this, from 620 human E3 ligases, we extracted amino acid sequences corresponding to 27 various human E3 domains (half of them were identified as catalytic) according to the InterPro database, crucial for E3 functioning [59]. Please note that several domains might be extracted from one unique E3. Next, we grouped 537 sequences of different domains for E3 ubiquitin ligases and their accessory proteins belonging to 283 proteins of the original sample. The consensus sequences of these clusters were used for alignment purposes. The total score values for clusters of E3 ligases and adaptor protein sequences can be found in Appendix A, respectively. Thus, the result of our study represents a homology of clusters (Figure 9B) and functional domains, representing groups of related clusters (Figure 9A,C), rather than a homology of E3 ligases and adaptor proteins itself. As E3 ligases have a modular structure (i.e., they may be assembled from different domain combinations), we think that this approach is more relevant to tracking its homology between different species. No homologues of were identified among Archaea and Bacteria, likely because key amino acids in Archaea and Bacteriaare of greater importance for enzyme function than the entire domain sequences.

A crucial set of E3 ligases was identified in all Eukaryotes, with the majority belonging to the RING ligase family, including the zf-rbx1-RING, znf-RING_MIZ, Cullin-NEDD8, and Znf_RING domains. Furthermore, in Metazoa, the total score values for these domains were above 50 (Figure 9A–C). The variation in clusters of E3 ligases and protein adaptors in different types of Metazoa was distributed the most divergently. The median values for total scores, presented as boxplots (Figure 9D), demonstrate that Echinodermata E3 are the closest to Chordata representatives. E3 of Mollusca, Arthropoda, Annelida, Cnidaria, and Rotifera have significantly lower homology, while median values for Nematoda and Platyhelminthes are close to zero. An analysis of E3 adaptors demonstrated an even more striking difference between Chordata and other phyla, where even Echinodermata E3 were strongly deviated from it. Other phyla with less than six individual annotated genomes were withdrawn from analysis.

In terms of their functional roles, E3 ligases (Figure 9B and Appendix A, group 3, Appendix A) containing these conserved clusters are involved in regulating several signal transduction pathways. These include the negative regulation of the MAPK signaling cascade, mTOR, Wnt, SREBP signaling, as well as the regulation of protein stability, positive regulation of protein catabolic processes, inflammatory response, and aspects of learning and memory (Figure 10).

In Chordata, Brachiopoda, Echinodermata, Priapulida, Annelida, Mollusca, Cnidaria, and Arthropoda, we can identify a crucial group 2 of homologues of human E3 ligases. This group includes the zf_C3HC4, Znf_RING, IBR, Sina, ATG5_UBLA, ATG5_HBR, RINGv, and U-box domains (Figure 9B and Appendix A, group 2). In contrast, Protista, Rhodophita, Ochrophyta, and Viridiplantae species possess only ATG5_UBLA, RINGv, and U-box domains, which are characterized by lower score values compared to Metazoa. Additionally, Placozoa, Porifera, Plathelminthes, and Nematoda species exhibited the presence of variable domains, including zf_C3HC4, Znf_RING, IBR, Sina, and ATG5_HBR. This variability is likely attributed to the parasitic lifestyle of many species within these groups.

The most conserved E3 domains among Metazoa and Viridipolantae (Figure 9B and Appendix A, group 5)—zf_C3HC4, zf-rbx1, ATG5_UblA, U-box, Cullin-NEDD8, RINGv, and zfRING—are functionally related to the regulation of various signaling cascades that support multicellularity. These E3 ligases also regulate cell differentiation, embryo and tissue specificity, and nervous system development, including neural tube formation and neuromuscular contacts (Figure 11, Appendix A).

The highly conserved Fbox-like, zf_C3HC4, and Zf-RING_LISH domains of E3 ligases were identified among Chordata, Brachiopoda, and Echinodermata types, while they exhibited greater variability in the Priapulida, Annelida, and Mollusca. The most specific and conserved domains for the Chordata type are VHL, Znf_B-box, IR1-M, SOCS-box, F-box, GIDE, and ATG16. In addition, Brachiopoda, Echinodermata, Priapulida, Annelida, Mollusca, Cnidaria, and Arthropoda species exhibited the presence of GIDE and ATG16 domains with lower total score values, indicating greater variability (Figure 9C). In terms of function, domains containing these clusters (Figure 9B and Appendix A, group 1) are associated with the processes that ensure the formation of a multicellular organism, including the development of individual organs, neural tube initiation, somite formation, development of the nervous system, neuronal migration during development, intrauterine development of the embryo, as well as such intracellular processes as maintenance of localization in the cell (Figure 12).

HECT ubiquitin ligases are thought to be highly conserved, while RING and RBR ubiquitin ligases show the greatest functional diversity. Our analysis revealed that among all Eukaryotes, the TRIAD and HECT catalytic domains of RBR and HECT E3 ligases exhibited the greatest variability (Figure 9A–C). This suggests that the effect of evolutionary selection to keep the amino acid sequence of the entire domain (TRIAD and HECT) may be weaker than that of individual amino acids within these domains, which are more important for their function than the domain itself. Consequently, the greater variability in the amino acid sequences of the TRIAD and HECT domains has likely resulted in a greater diversity of functions of RBR and HECT ligases. This was probably accompanied by a more complex life cycle and embryo- and tissue-specific expression of different RBR and HECT ligases. Functionally, RBR and HECT E3 ligases containing these domains (Figure 9B and Appendix A, group 4) are involved in the development of the nervous system, including brain development, corpus callosum formation, cerebellar Purkinje cell differentiation, regulation of synaptic plasticity, motor exploratory behavior, and motor learning (Figure 13). These E3 ligases also contribute to hematopoiesis, development of the immune response, regulation of cell communication, and signal transduction.

Except for APC2, every member of the cullin family is known to be modified by NEDD8 [112]. NEDD8 promotes dimerization and activation of cullin-RING E3 ligases [113,114,115]. It also acts as a linker that binds disparate elements of cullin and RING-activated ubiquitin-bound E2 [116]. In the case of *D. discoideum*, NEDDylation of cullins affects cell differentiation and enables the transition from a single-cell to a multicellular form of *D. discoideum’s* existence [117]. Regarding the adaptor proteins for E3 ligases, the NEDD8-binding domain within the cullin protein was found to be the most conserved among all Eukaryotes (Figure 9A–C). The other domains of Cullin1–5 exhibited considerable variability among all eukaryotic kingdoms, except for Parabasalia and Euglenozoa. The high variability of the cullin domain amino acid sequence is likely to permit more precise bilateral tuning of the interaction with proteins containing the RING domain and Skp1 adaptor proteins, which determine the interaction with substrates.

According to Gene Ontology biological pathways analysis, adaptor proteins and similar E3 ligases from group 1 (Figure 12) and group 5 (Figure 11) are functionally involved in the formation and development of organs of the nervous system, and many cellular processes, including cell division, cell population proliferation, cell motility, cellular response to stimulus, macromolecule metabolism, maintenance of multicellularity, maintenance of cell number, cell communication, as well as the positive and negative regulation of these processes (Figure 14, Appendix A).

Consequently, for all Eukaryotes, we can distinguish a set of the most conservative domains of E3 ligases, namely zf-rbx1, zf-RING, and Cullin-NEDD8 domains. These domains are responsible for performing basic cellular functions, whereas variable TRIAD and HECT domains, along with cullin adapter protein domains, are involved in more specific regulatory roles.

## 4. Conclusions

Since the discovery of the ubiquitination system involved in protein degradation in the 1980s, ubiquitin and Ubls have been found to directly or indirectly influence virtually all cellular processes with remarkable specificity. The evolution of Ubl proteins can be described as a process of diversification, which involves creating unique amino acid patterns, determining their specific functions, and maintaining the general composition of Ubls. This process was subject to purifying selection, a form of natural selection that acts to eliminate deleterious mutations. A set of conserved enzymes that ensure conjugation with the substrate accompanied the conservation of the Ub fold. At the same time, conservation of the E2 enzymes was observed from Archaea, while E1 and E3 exhibited greater variability in domain composition, probably for fine-tuning interactions with the conditionally novel Ubl, as well as for interactions with novel substrates.

The evolution of this system over 3.5 billion years was inevitably accompanied by the perfection of the enzymes involved in the conjugation of Ub and Ubls by increasing the complexity of their domain structure, which is a minimal evolutionary module [35]. This led to the formation of a group of highly conserved domains of eukaryotic ubiquitination enzymes, ensuring the regulation of basic cellular functions and maximizing the fine-tuning of key eukaryotic signaling cascades through the emergence of highly variable domains within these enzymes. Concurrently, the emergence of multicellularity was accompanied by the appearance of new clusters of conserved domains in E2 and E3 enzymes, which contributed to the fixation of their functions, ensuring multicellularity, differentiation, and development of organs and tissues. Here, we would like to emphasize that the main limitation of our study is that distinct ubiquitin ligases, which significantly differ from their human orthologues, may still exist in analyzed organisms and have the same functionality. Nevertheless, the provided global analyses should have enough power to discern evolutionary trends of the ubiquitination system, especially in the case of a statistically significant number of representatives.

The main driver of the ubiquitination system’s evolution remains enigmatic. The formation of new domains in the ubiquitination system enzymes at the stage of the LECA formation was probably due to HGT between Archaea and viruses, and at later stages to transfer between viruses and Eukaryotes. The mutation rate of viruses is known to be higher than that of Eukaryotes (~1.1 × 10^−8^ per site per generation in humans [118]), ranging from 10^−3^ to 10^−6^ copy errors per nucleotide in RNA viruses [119] and 10^−6^ to 10^−8^ mutations per base pair per generation in DNA-containing viruses [120]. Recently, Irwin and Richards have demonstrated that viruses played a role in the evolution of nucleosomes. Viral histone paralogues occupy an intermediate position between histones of Archaea and early Eukaryotes and can participate in the assembly of complexes that resemble both eukaryotic nucleosomes and Archaea. These complexes also have the capacity to influence genomic activity and condense DNA [121]. Thus, the incredible diversity of ubiquitin ligases may, at least in part, be driven by viral mutagenesis drift with beneficial variants secured by evolutionary pressure.

## Figures and Tables

**Figure 1 ijms-25-08671-f001:**
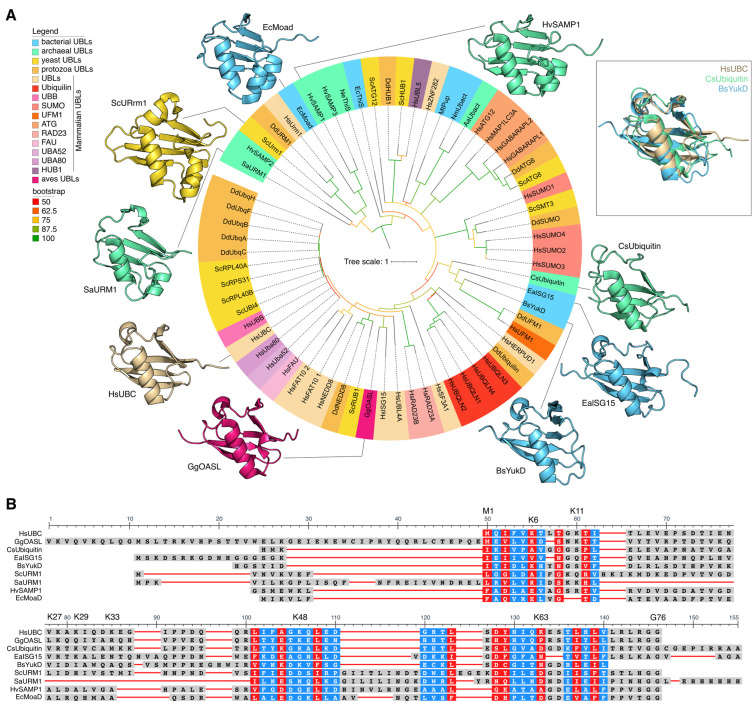
(**A**) Phylogenetic tree based on amino acid sequences of ubiquitin-like domains. Proteins from the following groups of organisms are shown: mammals (*Hs*—*Homo Sapiens*), birds (*Gg*—*Gallus gallus*), yeast (*Sc*—*Saccharomyces cerevisiae*), Protozoa (*Dd*—*Dictyostelium discoideum*), bacteria (*Bs*—*Bacillus subtilis*, *Ea*—*Ensifer aridi*, *Ec*—*Escherichia coli*, *Mt*—*Mycobacterium tuberculosis*, *Nm*—*Nitrospira moscoviensis*), Archaea (*Aa—Candidatus Aenigmarchaeota Archaeon*, *Cs*—*Caldiarchaeum subterraneum*, *Hv*—*Haloferax volcanii*, *Ne*—*Nanoarchaeum equitans*, *Sa*—*Sulfolobus acidocaldarius*) (Appendix A). Clades with bootstrap ≥ 50 were considered reliable. (**B**) The structural alignment of human ubiquitin, archaeal ubiquitin, and bacterial ubiquitin-like protein. Multiple amino acid sequence alignment of the ubiquitin-like domains of HsUBC, GgOASL, CsUbiquitin, EaISG15, BsYukD, ScURM1, SaURM1, HvSAMP1, and EcMoaD proteins. Highly conservative positions are highlighted in red; less-conservative positions are highlighted in blue.

**Figure 2 ijms-25-08671-f002:**
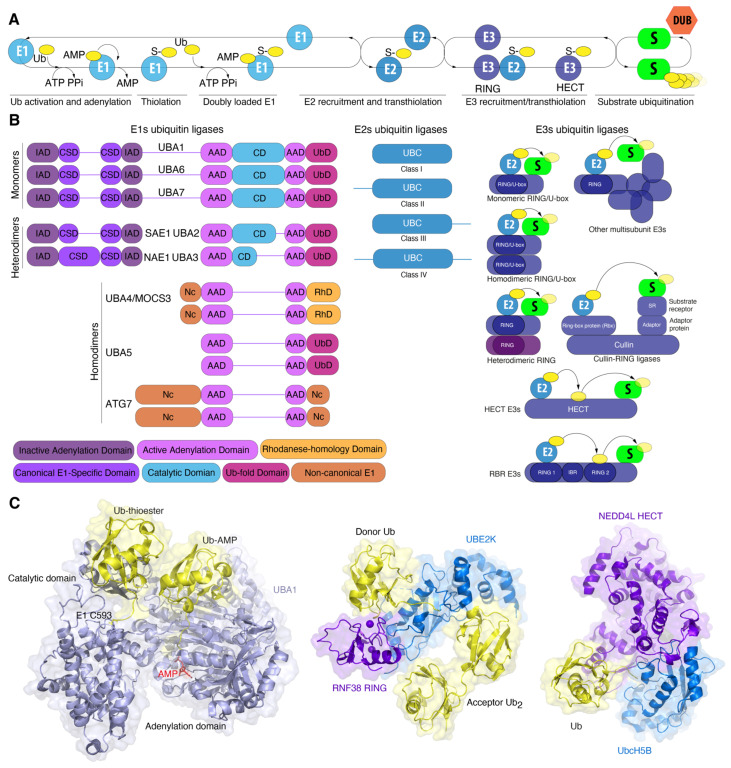
Ubiquitin–substrate conjugation cascade (**A**) and domain organization of human E1-E2-E3 enzymes (**B**). E1 domains are labelled and colored according to the legend (bottom). Please refer to main text for details. (**C**) Crystal structure of E1 UBA1 (light blue) simultaneously bound to two Ub molecules (yellow): non-covalently bound Ub-AMP in the active adenylation domain, and covalently bound Ub–thioester (left, PDB: 4NNJ). Crystal structure of complex of E2 UBE2K and E3 RING of human RNF38 ligase (central, PDB: 7OJX). Donor ubiquitin is covalently bound to E2 UBE2K. Crystal structure of the E3 ligase of human NEDD4L HECT bound to Ub and the E2 conjugating enzyme UbcH5B (right, PDB: 3JW0).

**Figure 3 ijms-25-08671-f003:**
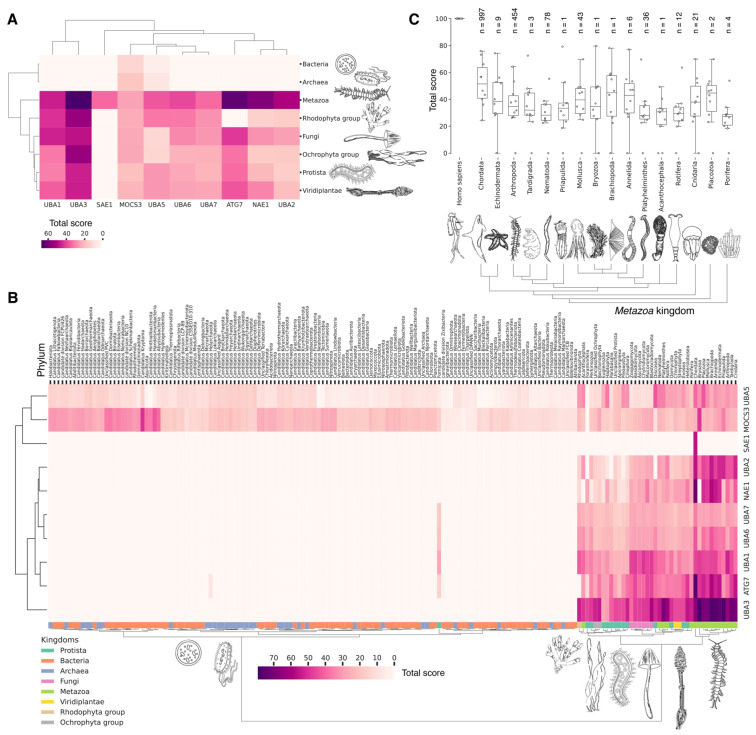
Hierarchically clustered heat map with total score values for E1 activating enzymes classified by phyla (**A**) and kingdoms (**B**). (**C**) Boxplots by individual total score values of E1 activating enzymes for the Metazoa kingdom. Each dot represents average value for unique E1 among analyzed phylum representatives. Normalized values (0–100) characterize the proximity of catalytic domains of E1 ubiquitin ligases to human representatives. For each phylum, the median and interquartile values are given. Bars represent standard deviation values. Count of individual analyzed genomes is indicated (*n*).

**Figure 4 ijms-25-08671-f004:**
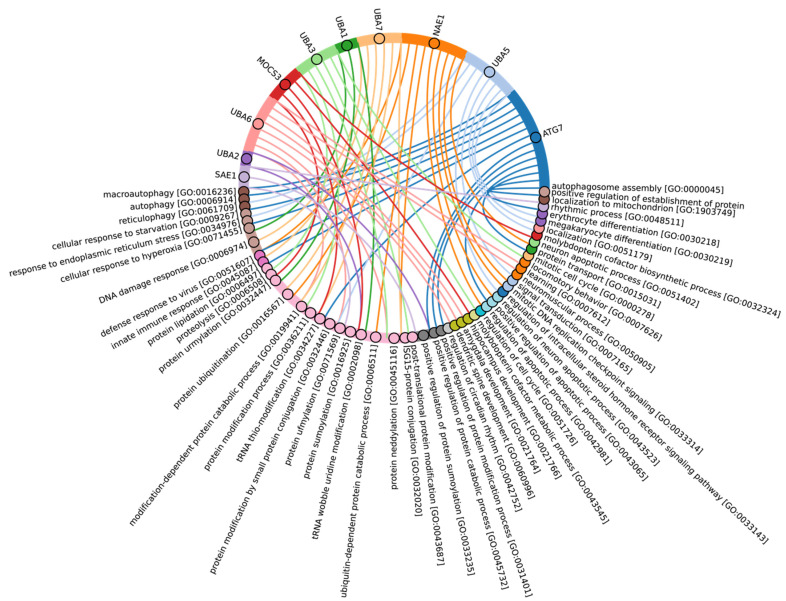
Chord diagram for human E1 enzymes showing the relationship of the proteins to the Gene Ontology biological pathways in which these proteins are involved.

**Figure 5 ijms-25-08671-f005:**
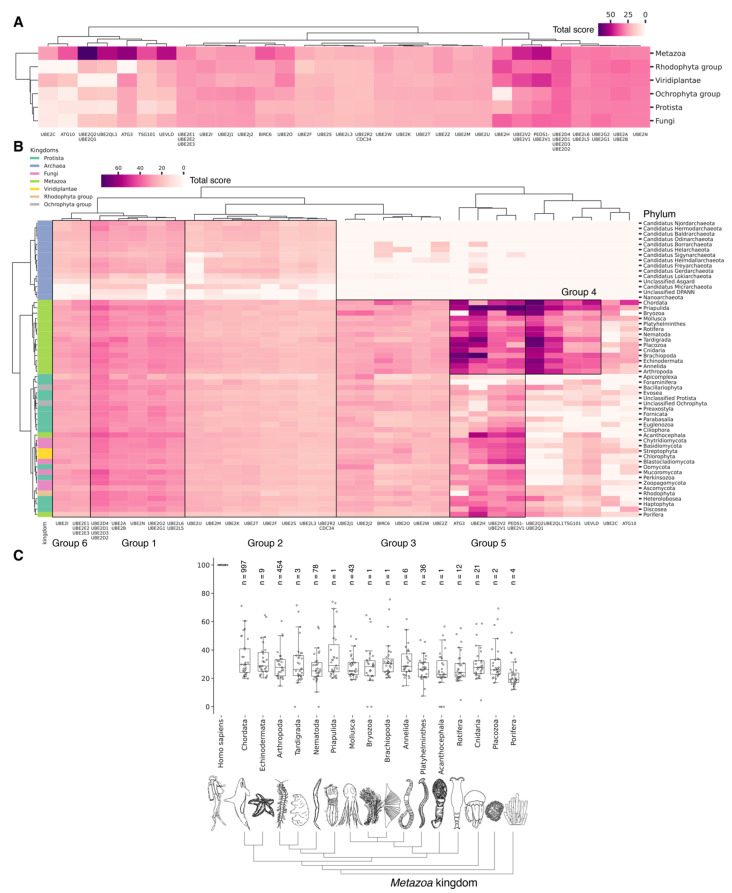
Hierarchically clustered heat map with total score values for E2 activating enzymes classified by phyla (**A**) and kingdoms (**B**) with indication of E2 group boundaries for functional analysis using the Gene Ontology database (https://geneontology.org/, (accessed on 3 May 2024) [89]), (Appendix A). (**C**) Boxplots by total score values of E2 activating enzymes for Metazoa kingdoms. Each dot represents average value for unique E2 (or group of highly homological E2) among analyzed phylum representatives. Normalized values (0–100) characterize the proximity of catalytic domains of E2 ubiquitin ligases to human representatives. For each phylum, the median and interquartile values are given. Bars represent standard deviations. Count of individual analyzed genomes is indicated (*n*).

**Figure 6 ijms-25-08671-f006:**
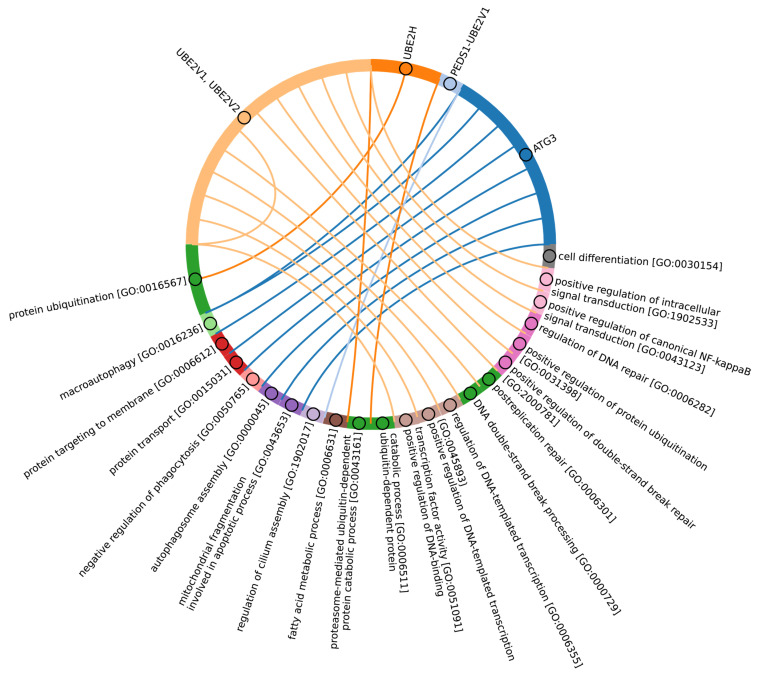
Chord diagram illustrating the relationship between the E2 enzyme group 5 proteins and their associated Gene Ontology biological pathways (Appendix A).

**Figure 7 ijms-25-08671-f007:**
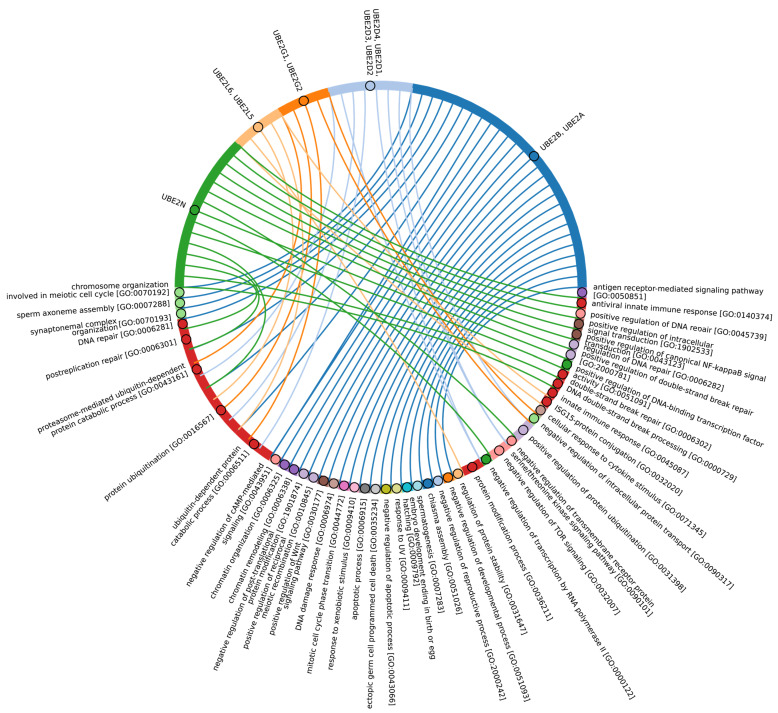
Chord diagram illustrating the relationship between the E2 enzyme group 1 and their associated Gene Ontology biological pathways (Appendix A).

**Figure 8 ijms-25-08671-f008:**
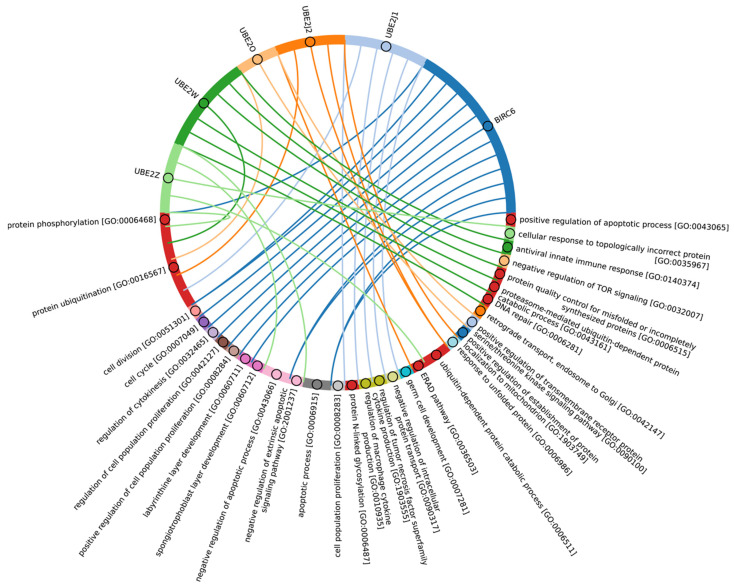
Chord diagram illustrating the relationship between the E2 enzyme group 3 and their associated Gene Ontology biological pathways (Appendix A).

**Figure 9 ijms-25-08671-f009:**
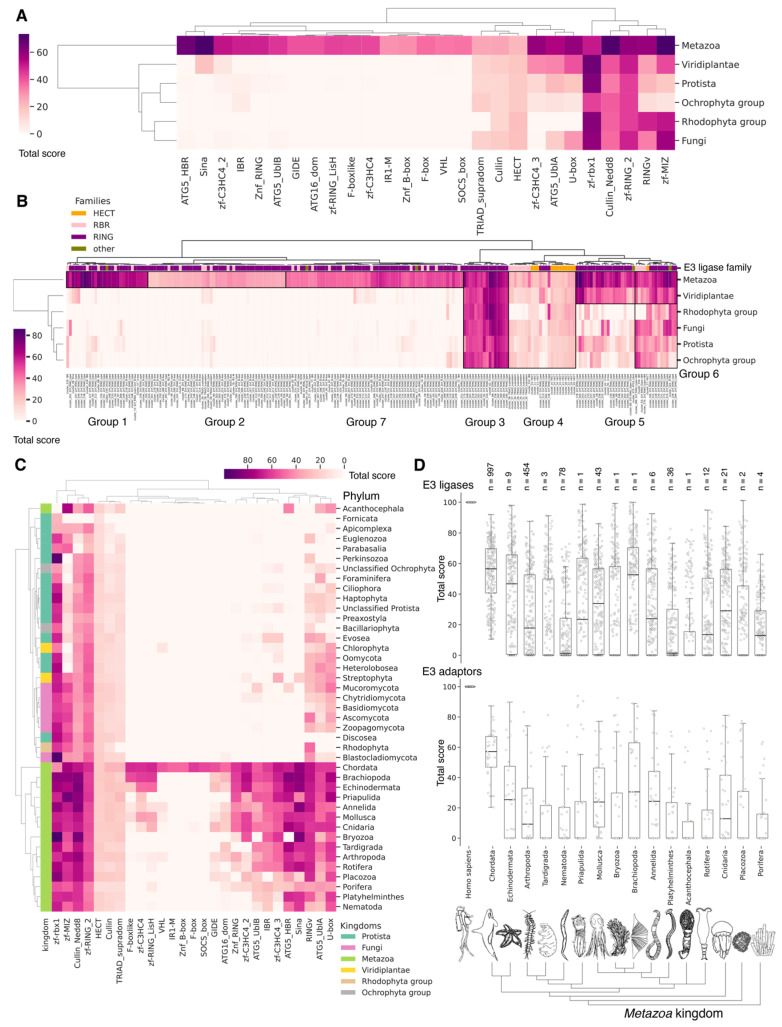
Hierarchically clustered heat map with total score values for domains (**A**,**C**) and clusters (**B**) of E3 ligases classified by phyla (**A**,**B**) and kingdoms (**C**). Groups marked on panel (**B**) indicate E3 boundaries to assess the relationship to Gene Ontology biological pathways in which proteins containing these clusters are involved. In panels (**A**,**C**), we used median of total scores for groups of clusters belonging to distinct domains. In panel (**B**) and Appendix A, the total scores for clusters of E3 ligases and adapter proteins were used. Normalized values (0–100) characterize the proximity of functional domains or clusters of E3 ubiquitin ligases to human representatives. (**D**) Boxplots by total score values of E3 ligases and adapter proteins clusters for Metazoa. Each dot represents average value for unique cluster among analyzed phylum representatives. For each phylum, the median and interquartile values are given. Bars represent standard deviations. Count of individual analyzed genomes is indicated (*n*).

**Figure 10 ijms-25-08671-f010:**
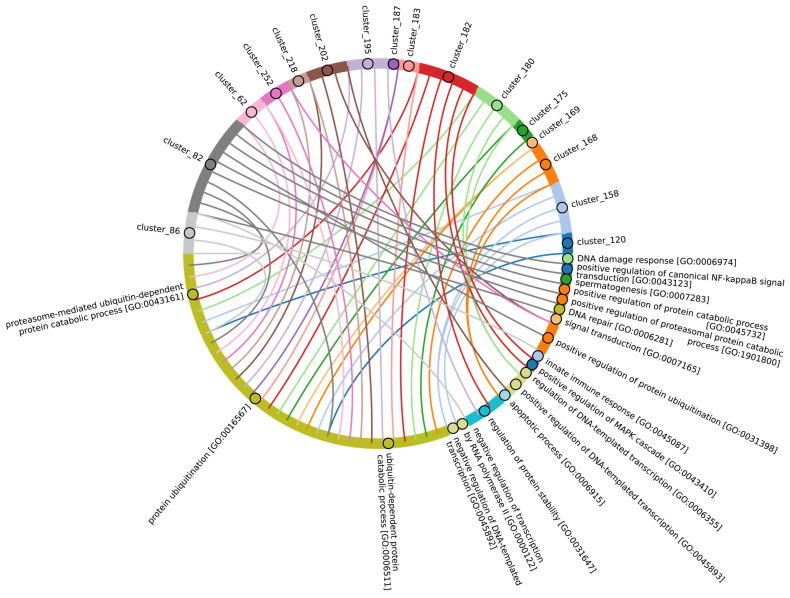
Chord diagram illustrating the relationship between the E3 ligase containing clusters from group 3, and their associated Gene Ontology biological pathways. The pathways associated with at least 15 proteins in the total sample of E3 ligases are shown. The complete (unabbreviated) list of pathways can be found in the Appendix A.

**Figure 11 ijms-25-08671-f011:**
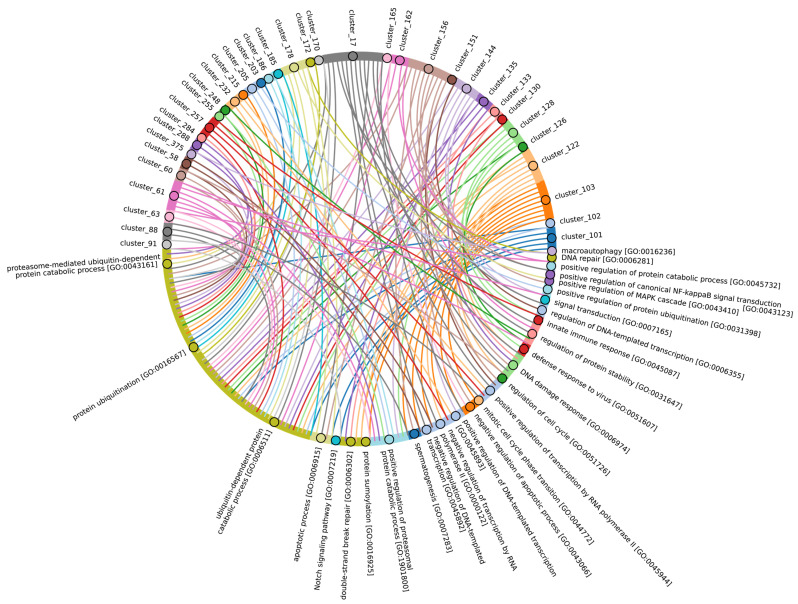
Chord diagram illustrating the relationship between the E3 ligase containing clusters from group 5, and their associated Gene Ontology biological pathways. The pathways associated with at least 15 proteins in the total sample of E3 ligases are shown. The complete (unabbreviated) list of pathways can be found in the Appendix A.

**Figure 12 ijms-25-08671-f012:**
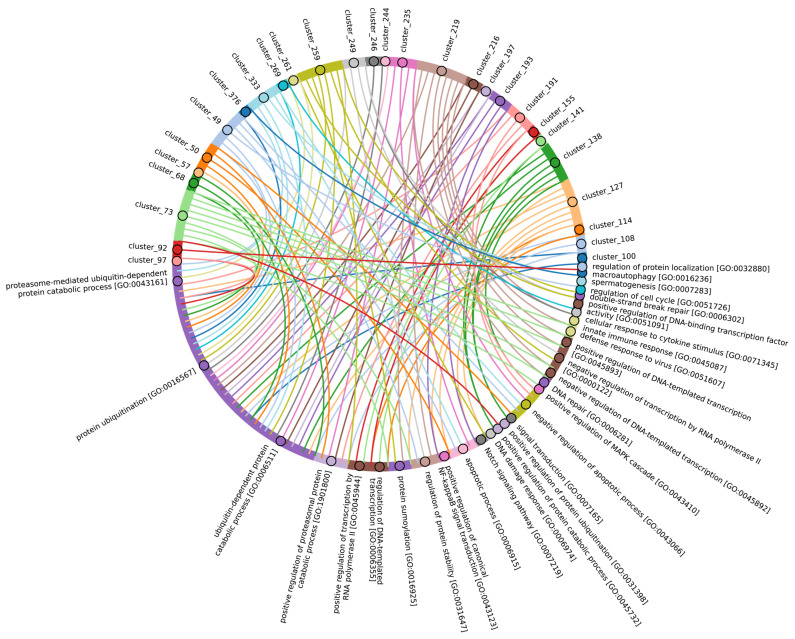
Chord diagram illustrating the relationship between the E3 ligase containing clusters from group 1, and their associated Gene Ontology biological pathways. The pathways associated with at least 15 proteins in the total sample of E3 ligases are shown. The complete (unabbreviated) list of pathways can be found in the Appendix A.

**Figure 13 ijms-25-08671-f013:**
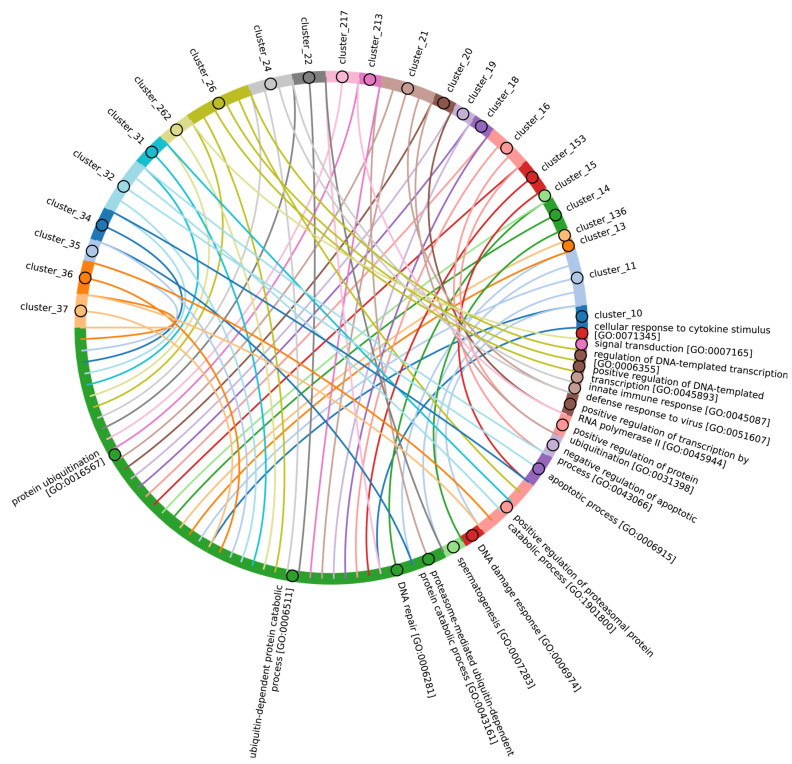
Chord diagram illustrating the relationship between the E3 ligase containing clusters from group 4, and their associated Gene Ontology biological pathways. The pathways associated with at least 15 proteins in the total sample of E3 ligases are shown. The complete (unabbreviated) list of pathways can be found in the Appendix A.

**Figure 14 ijms-25-08671-f014:**
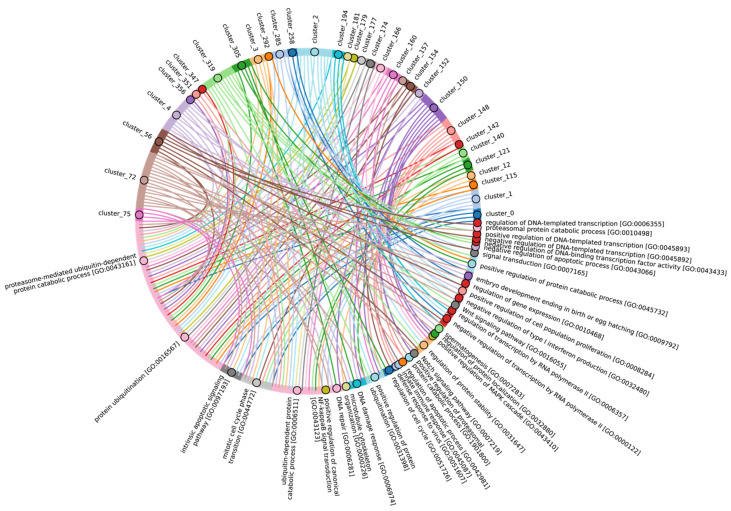
Chord diagram illustrating the relationship between the E3 adaptor proteins, containing clusters from groups 1 and 5, and their associated Gene Ontology biological pathways. The pathways associated with at least 10 proteins in the total sample of E3 ligases are shown. The complete (unabbreviated) list of pathways can be found in the Appendix A.

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
