# Peer review of "Tracking of Ubiquitin Signaling through 3.5 Billion Years of Combinatorial Conjugation"

_ijms, 2024, doi:10.3390/ijms25168671_

Round 1

Reviewer 1 Report

Comments and Suggestions for Authors

This is a well-reasoned and supported review of the evolution of the ubiquitination system across the domains of life.  I have one minor concern that I hope the authors will accept.  The use of the terms "higher" eukaryotes and "decisive advantage" to chordates in the abstract should be changed.  As written they support that notion that evolution proceeds by increasing the complexity of organisms (progressive evolution).  This idea is wrong.  "Higher" eukaryotes could be changed to "multicellular" without changing the authors indent. The diversification of ubiquitination function in eukaryotes, certainly explains how the systems evolved, but promoting the idea of a decisive advantage to chordates is also wrong, as the most successful metazoans on the planet are arthropods (and some might argue nematodes) that don't have the same system as mentioned by the authors.  Again this can be easily fixed by simply saying "which plays a crucial role in Chordate evolution."

Author Response

This is a well-reasoned and supported review of the evolution of the
ubiquitination system across the domains of life.  I have one minor
concern that I hope the authors will accept. The use of the terms "higher" eukaryotes and "decisive advantage" to chordates in the abstract should be changed. As written they support that notion that evolution proceeds by increasing the complexity of organisms (progressive evolution).  This idea is wrong.  "Higher" eukaryotes could be changed to "multicellular" without changing the authors indent. The diversification of ubiquitination function in eukaryotes, certainly explains how the systems evolved, but promoting the idea of a decisive advantage to chordates is also wrong, as the most
successful metazoans on the planet are arthropods (and some might
argue nematodes) that don't have the same system as mentioned by the
authors.  Again this can be easily fixed by simply saying "which plays
a crucial role in Chordate evolution."

We thank the Reviewer for this very valuable comments.

In response to her/his suggestions, we have made the following changes to our manuscript:

On line 15, we have replaced the term "Higher" eukaryotes with "multicellular" to provide a more accurate description.

On lines 29-30, we have revised the phrase “which delivered a decisive advantage to Chordata” to “which plays a crucial role in Chordate evolution” to better reflect the importance of our findings in evolutionary terms.

Reviewer 2 Report

Comments and Suggestions for Authors

Among post-translational modification ubiquitination offers a broad array of complexity. Though, ubiquitination was initially considered featuring eukaryotic cells, rather recently we have learned that ancestral ubiquitination systems exist also in Archeabacteria.

In the manuscript titled "Tracking of Ubiquitin Signaling Through 3.5 Billion Years of Combinatorial Conjugation" Kaminskaya A. and colleagues attempted to examine the diversity of the ubiquitination system among different life kingdoms.

The starting purpose is excellent but the outcome is less thrilling.

Please find below my concerns.

  • The major weakness is the rationale behind the analysis. Indeed, for their inquiry, the authors used only the catalytic domains of E1, E2, and E3. The protein domains represent the unit of modular evolution. Though their functional importance has been widely recognized in the UPS field in the present survey they have not been considered at all. On the contrary, they are crucial, especially for the E3 members. Basically, the reasons for their importance are two. First, they participate in discriminating, selectively, the broad array of the substrates. Second, it has to be kept in mind that often E3 members face a serious conundrum which is the ability to recognize the unfolded substrates. More importantly, often protein domain organization is not restricted to structural domains but it seems that, throughout evolution, more that structural commonalities are important the functional features. All these issues should have been considered and not omitted when planning a survey like the one the authors carried-out.
  • The main text should be profundly edited because to many extents looks verbose, thus losing its effectiveness. As a paradigmatic example the repetitions (e.g. lines 465-466 and 516-518 convey, basically, the same message). By trimming the main text the manuscript will be benefited
  • References are not formatted according to the journal guidelines
  • The section "Supplementary Methods" should be implemented. For instance: how the authors drew the heatmaps and chord diagrams should be detailed (e.g. which program did they use? R? Python? Else? Where did the bit-scores come from?).
  • Acronyms should be spelled out completely on initial appearance in text.
  • The labeling of the horizontal axis of Figures 3B and even more 9B is almost useless, due to the very small font. As such they are not legible, thus losing their meaning.
  • It seems that a couple of notional mistakes are present. To the reviewer's knowledge, currently, the recognized kingdoms are six: a) Archebacteria; b) Eubacteria; c) Protista; d) Fungi; e) Plantae, and f) Animalia. Viruses are not considered because by lacking their own metabolism they are unable to grow and replicate by themselves. Additionally, quite often the authors misused the term Phylum instead of Genus.
  • A few typos are scattered throughout the main text (conventionally the genus should be in italics font -i.e., lines 493-495-)
  • The size of the different metazoan subgroups used to depict the total score (e.g. Figures 3C, 5C, etc…) is rather heterogeneous ranging from n=997 to n=1. Such heterogeneity makes very difficult to draw robust conclusions, is it not?
Comments on the Quality of English Language

The English language would benefit from a trimming and careful editing, because as such looks verbose. Overall, the latter affects on the effectiveness of the messages conveyed.

Author Response

Among post-translational modification ubiquitination offers a broad array of complexity. Though, ubiquitination was initially considered featuring eukaryotic cells, rather recently we have learned that ancestral ubiquitination systems exist also in Archeabacteria. In the manuscript titled "Tracking of Ubiquitin Signaling Through 3.5 Billion Years of Combinatorial Conjugation" Kaminskaya A. and colleagues attempted to examine the diversity of the ubiquitination system among different life kingdoms.

The starting purpose is excellent but the outcome is less thrilling.

Please find below my concerns.

The major weakness is the rationale behind the analysis. Indeed, for their inquiry, the authors used only the catalytic domains of E1, E2, and E3. The protein domains represent the unit of modular evolution. Though their functional importance has been widely recognized in the UPS field in the present survey they have not been considered at all. On the contrary, they are crucial, especially for the E3 members. Basically, the reasons for their importance are two. First, they participate in discriminating, selectively, the broad array of the substrates. Second, it has to be kept in mind that often E3 members face a serious conundrum which is the ability to recognize the unfolded substrates. More importantly, often protein domain organization is not restricted to structural domains but it seems that, throughout evolution, more that structural commonalities are important the functional features. All these issues should have been considered and not omitted when planning a survey like the one the authors carried-out.

We thank the Reviewer for her/his valuable comments. We fully acknowledge the Reviewer’s criticism and would like to apologize for the term “catalytic domains”, especially in case of E3 ligases, because it was a typo mistake. Please follow our statements, which are now included in the main text.

  1. In case of E1 analysis of catalytic domains but not full sequences was reasoned by high degree of false-positive homology occurred due to the significantly increased frequency of other domains, e.g. adenylation domain, in proteins completely not related to ubiquitin ligases. Here true “catalytic domains”.

  1. As amino acid sequence of E2 ligases consists mainly of UBC domain with short N- or C-terminal extensions (Figure 2B), we examined the homology of the catalytic UBC domains of E2 conjugating enzymes (generally representing almost full protein sequence) across Eukaryotes and Prokaryotes. Here aging “true catalytic”, covering majority of full sequence.

  1. In case of E3 ligases and adaptors, assuming its variability, from 620 human E3 ligases we extracted amino acid sequences corresponded to 27 various human E3 domains (only half of them were identified as catalytic) according to InterPro database, crucial for E3 functioning [Ref. 59]. Please note, that several domains might be extracted from one unique E3. Next, we grouped 537 sequences of different domains of E3 ubiquitin ligases and their accessory proteins belonging to 283 proteins of the original sample. The consensus sequences of these clusters were used for alignment purposes. The total score values for clusters of E3 ligases and adaptor proteins sequences can be found in Supplementary Tables S6_1 and S6_2, respectively. Thus, result of our study represents homology of clusters (Figure 9B) and functional domains, representing group of related clusters (Figure 9A and 9C), rather than homology of E3 ligases and adaptor proteins itself. As E3 ligases have modular structure (i.e. they may be assembled from different domain combinations), we think that this approach is more relevant to track its homology between different species.

Legend to Figure 9 now reads

Figure 9. Hierarchically clustered heat map with total score values for domains (A,C) and clusters (B) of E3 ligases classified by phyla (A,B) and kingdoms (C). Groups marked on panel (B) indicates E3 boundaries to assess the relationship to Gene Ontology biological pathways in which proteins, containing these clusters, are involved. In panels 9A and 9C we used median of total score for group of clusters belonging to distinct domain. In panel 9B and Figure S3 the total scores for clusters of E3 ligases and adapter proteins were used. (D) Boxplots by total score values of E3 ligases and adapter proteins clusters for Metazoa. Each dot represents average value for unique cluster among analyzed phylum representatives. For each phylum, the median and interquartile values are given. Bars represent standard deviations. Count of individual analyzed genomes is indicated (n).

We again apologize for this misunderstanding. All mentioning of “catalytic domains” were cleared in E3 chapter.

The main text should be profundly edited because to many extents looks
verbose, thus losing its effectiveness. As a paradigmatic example the
repetitions (e.g. lines 465-466 and 516-518 convey, basically, the
same message). By trimming the main text the manuscript will be
benefited

In response to Reviewer’s suggestion, main text was carefully proof-red. The English in lines 465-466 and 516-518 has been improved for greater clarity and readability.

References are not formatted according to the journal guidelines

The references have been reformatted to align with the journal's guidelines.

The section "Supplementary Methods" should be implemented. For
instance: how the authors drew the heatmaps and chord diagrams should
be detailed (e.g. which program did they use? R? Python? Else? Where
did the bit-scores come from?).

The bit-score metric, which appears in the Blast search results tables, is used to estimate sequence homology. As specified in the Methods section:

"The search produced a set of homologous proteins with an e-value threshold of 1e-4 for each original sequence (consensus or domain). The bit-score was utilized as a measure of sequence homology."

Normalized values (0-100) characterize the proximity of ubiquitin ligases (catalytic domains, clusters, functional domains) to human representatives. Each dot on boxplot represents average value for unique E1, E2 (or group of highly homological E2) or among analyzed phylum representatives. Figure legends were modified accordingly.

Additionally, the Methods section has been updated to indicate the specific programs used for generating graphs.

Acronyms should be spelled out completely on initial appearance in text.

Acronyms have been fully spelled out.

The labeling of the horizontal axis of Figures 3B and even more 9B is
almost useless, due to the very small font. As such they are not
legible, thus losing their meaning.

Unfortunately, due to the large number of taxonomic groups analyzed, it is not possible to increase the size of the captions in the figures within the article. However, full-size versions of these figures have been added to the Supplementary materials.

It seems that a couple of notional mistakes are present. To the
reviewer's knowledge, currently, the recognized kingdoms are six: a)
Archebacteria; b) Eubacteria; c) Protista; d) Fungi; e) Plantae, and
f) Animalia. Viruses are not considered because by lacking their own
metabolism they are unable to grow and replicate by themselves.

In our study, we utilized the NCBI taxonomic classification (please refer to https://www.ncbi.nlm.nih.gov/pmc/articles/PMC3245000/ and https://www.ncbi.nlm.nih.gov/pmc/articles/PMC7408187/), from which we derived the names of taxonomic ranks and groups. However, since some species lack confirmed classification within established primary ranks such as kingdom or phylum, we made several adjustments to the main ranks of these species solely for the sake of data presentation. All these adjustments are detailed in the Methods section.

Additionally, quite often the authors misused the term Phylum instead
of Genus.

We did not examine organisms at the genus level. We have strictly adhered to the NCBI taxonomic classification system, which ensures that all terms are used correctly and consistently, with no misclassification.

A few typos are scattered throughout the main text (conventionally the
genus should be in italics font -i.e., lines 493-495-)

These errors have been corrected to ensure proper formatting and consistency.

The size of the different metazoan subgroups used to depict the total
score (e.g. Figures 3C, 5C, etc…) is rather heterogeneous ranging from
n=997 to n=1. Such heterogeneity makes very difficult to draw robust
conclusions, is it not?

Indeed, yes. Please note, that we discuss only phyla with more than 6 individual representatives.

Figures and main text were modified accordingly.

Phyla with less than 6 individual annotated genomes were withdrew from analysis.

Count of individual analyzed genomes is indicated (n).

Round 2

Reviewer 2 Report

Comments and Suggestions for Authors

The reviewer appreciated the articulated response drafted by the authors. Though, some issues remain still unanswered (e.g., references formatting -lines 382, 387, 495, 599, etc…-) most of the original concerns have been clarified in the revised version of the manuscript. Nonetheless, before publication the authors are warmly advised to edit and proofread the manuscript once more.

Author Response

We thank the Reviewer for careful consideration and the valuable comments provided.

We fixed the remaining issues, including the formatting of references on lines 382, 387, 495, 599, and others. Additionally, we have meticulously edited and proofread the entire manuscript to ensure that all issued have been resolved.